# Patterns of fear of crime: A mixed-methods exploration of the model of experiential and expressive fear of crime

**Julien Noble** ⓘ *◉, **Antoine Jardin**◉

Centre de Recherches Sociologiques sur le Droit et les Institutions Pénales (CESDIP), Centre National de la Recherche Scientifique (UMR 8183), Guyancourt, France

◉ These authors contributed equally to this work.
* julien.noble@cesdip.fr

## Abstract

Hundreds of studies have been published on fear of crime, but few models offer a unified framework for this social phenomenon. This is the ambition of the model of experiential and expressive fear of crime (EEF) developed in the mid-2000s by a team of British researchers. However, despite its numerous contributions, this original model faces two limitations. First, the different versions of the model present certain discrepancies. Second, the statistical methods used consistently rely on hypothetico-deductive reasoning, which may overlook the existence of certain combinations of variables. The objective of this article, based on a French piece of research, is to revisit this model by combining two different families of statistical methods: multivariate configuration analysis and logistic regression analysis. The first analysis attempts to overcome these two limitations by adopting an inductive approach, which involves studying the multidimensional structure of the data without imposing pre-defined structures on them. It identifies four classes of respondents, each associated with a specific relationship to experiential fear and expressive fear. If the 'unworried' and the 'worried-dysfunctional' associate these two dimensions of fear of crime, the 'anxious' and the 'worried-functional' clearly separate them. By adopting an inferential approach, the second analysis aims to determine the sociodemographic factors of these different groups. It shows that the predictors vary significantly from one group to another and that no variable (not even gender) is a predictor for all groups. These results encourage moving beyond the dichotomous conceptualization ('worried'/'unworried'), which is still widely used in the study of fear of crime. Systematically identifying these different groups could also help combat fear of crime more effectively by implementing targeted and adapted public policies.

**Data availability statement:** All relevant data are within the paper and its Supporting Information files. The raw data may be accessed at https://data.sciencespo.fr/data-set.xhtml?persistentId=doi:10.21410/7E4/VD9RGA.

**Funding:** This work was funded by the Maison des Sciences de l'Homme Paris-Saclay - Emergence Project 2021: https://msh-paris-saclay.fr/spip2022/ Initials of the authors who received each award: JN et AJ Grant numbers awarded to each author: N/A The full name of each funder : Maison des Sciences de l'Homme Paris-Saclay - Emergence Project 2021 URL of each funder website: https://msh-paris-saclay.fr/spip2022/ MSH Paris Saclay has no role in the in the study design, data collection and analysis, decision to publish, or preparation of the manuscript.

**Competing interests:** The authors have declared that no competing interests exist.

## Introduction

Between the mid-2000s and the mid-2010s, a team of British researchers developed the model of experiential and expressive fear of crime [1–4]. The authors' ambition is to put forward an unified explanatory model of this social phenomenon. Their goal is to integrate the knowledge about fear of crime into a single model; in other words, to coherently and intelligibly combine the specific contributions of studies focused on particular aspects of the phenomenon. This aim to 'unify' the knowledge is not a new development. The history of research on fear of crime includes several similar attempts [5–14]. However, the model proposed by Farrall, Jackson, and their colleagues has the advantage of being based on a very recent conceptualization and operationalization of fear of crime. Also, it integrates, from an updated literature review, the factors recognized by previous studies as being the most contributive to this social phenomenon. The originality of the model lies in the differentiation between two dimensions of fear of crime. The researchers differentiate between concrete episodes of worry about crime in perceived threat situations (experiential fear) and more abstract and diffuse anxiety concerning crime as a societal problem (expressive fear).

This model has had a significant impact on scientific literature on fear of crime and is currently considered a key reference for recent research [15–24]. However, with the exception of the analyses conducted by Farrall, Jackson, Gray, and Hirtenlehner, the model of experiential and expressive fear of crime (EEF) has been very scantily tested empirically. This task seems all the more necessary as there are currently three versions of the EEF model, which, in some respects, are difficult to reconcile. The first [1,4] and third versions [3], developed using structural equation models (SEM), strongly associate expressive fear and experiential fear, with the second dimension being heavily influenced by the first. Developed from a theoretical typology supported by regression analyses, the second version reveals a more complex and nuanced relationship between the two dimensions of fear of crime [2].

In the current state of knowledge, the EEF model faces two main limitations. First, the three versions of the model show discrepancies in the relationship between certain variables. As we will see, these discrepancies tend to weaken the initial premise of the model – largely defended in Jackson [4] and Farrall et al. [1] – that expressive fear indirectly influences experiential fear through perceptions of the neighborhood. Second, the statistical methods used consistently rely on hypothetico-deductive reasoning. Whether using structural equation models (SEM) or theoretically designed typologies, these methods share the common approach of determining *a priori* the structure of relationships best suited to the data to explain the phenomenon studied. However, given the large number of variables involved in the EEF model, constructing models based on hypothetico-deductive reasoning significantly increases the risk of missing the existence of other combinations of variables (which researchers might not have considered *a priori*), thus rendering original relationships between the two dimensions of the fear of crime invisible [25–29].

This article seeks to answer the following question: Have the different versions of the EEF model succeeded in identifying all the relationship structures between the variables integrated into the model?

Based on a survey conducted in 2021 with a representative sample of the French population, this article aims to revisit the EEF model using different families of statistical methods. Its objective is twofold:

- Using a multivariate configuration analysis, we will first seek to identify the interactions that are involved in the construction of the structures of the relationship to experiential and expressive fear of crime.

- Using a logistic regression analysis, we will then seek to determine the sociodemographic predictors of each of the classes identified in the multivariate configuration analysis.

## Theoretical framework

To attempt to demonstrate the contribution of this methodological approach, it is essential to first present the composition, functioning, and results of the different versions of the EEF model.

- From the model's origins to its design

The EEF model is a continuation of Ferraro's work. In the mid-1990s, this researcher introduced a paradigm shift in the study of fear of crime. Until then, most research adopted a risk factor approach. This consists of identifying the population groups that are the most susceptible to fear of crime because of their individual and socio-demographic characteristics [For a review of the literature on these many studies, please see Farrall et al. [1], Hale [30], Lane et al. [21], or Vanderveen [31]]. Ferraro [7] was one of the first researchers to reverse this approach by proposing a model for the *interpretation of risk*. He considered it was not sufficient merely to identify the population groups that are the most susceptible to fear of crime or even the most conducive contexts for its emergence. Instead he thought it necessary to explain the psychological and social processes that underpin this phenomenon. More precisely, this author attempted to account for the mechanisms by which the evaluation of the environment provides information which can then be used to construct an estimate of the risk of victimization. The model's first contribution is to provide a central place to the perception of the risk of victimization, the predominant role of which has already been highlighted by several previous works [13,14,32–37]. Ferraro considers that fear of crime is not directly determined by a neighborhood's social and environmental characteristics or even by individuals' characteristics (objective situations) but by the interpretation that individuals make of these characteristics in the situation (definition of the situation). In this sense, the perception of the risk of victimization plays a central mediating role between local conditions and fear of crime. The model's second positive contribution is that it showed that the perception of incivility influences the evaluation of risk more than the presence or concentration of such incivility. To increase the perception of the risk of victimization, environmental cues (physical and social incivility) need to be interpreted as 'signs' of danger, crime or disorder that are often associated with the idea of a decline in informal social control and social cohesion within the community. Later, this line of research was taken further by Innes [38,39], Innes et al. [40], Sampson et Raudenbush [41,42] and Sampson [43]. The third positive contribution made by Ferraro's model is that it showed the diversity of reactions to a perceived risk. Fear is only one of these reactions. The resources available to individuals and how they define the situation at hand may lead them to adopt avoidance and/or defensive behavior [44–49] or become involved in political or community activism [50,51].

Farrall, Jackson and Gray's model takes Ferraro's one further in two directions. Firstly upstream, by attempting to identify the factors that are part of the influences on the perception of disorder as threatening. Secondly downstream, by proposing a more recent conceptualization and operationalization of fear of crime.

The EEF model aims to find answers to the following upstream question – what shapes how people evaluate disorder above and beyond neighborhood's characteristics? In other words, how can we explain that one individual may judge an ambiguous *stimulus* to be threatening while another person living in the same environment judges it benign and harmless

[4,8,22,23,27,43]? Farrall et al. [1] considered the answer to lie in forms of judgments, appreciations and attitudes that are closely associated with fear of crime. Their hypothesis on this point was primarily based on a series of qualitative studies carried out in the 1990s, the results of which show how respondents' comments about crime tended to be structured around other forms of concern about social, economic and political order. Anxiety about crime and disorder acts as a 'pointer' to many other issues going much further than just acts of crime to include the quality of social ties [52–55], trust in the community [56], cultural and racial stereotypes in the residents' neighborhoods [57–59], and more broadly concern about the evolution of social relations [60,61], respect for moral and political values [61–63] and society as a whole [64]. Several quantitative studies subsequently corroborated these results. Collective efficacy was said to be associated with a lower perception of disorder [4] and a lower perception of the risk of victimization [65]. Conversely, authoritarian attitudes and the most 'declinist' standpoints regarding the community in the long term were thought to favor the perception of disorder [4]. Cultural and racial stereotypes were also thought to reinforce the evaluation of the environment as being threatening, particularly in urban ghettos, by strongly associating crime with disadvantaged ethnic populations [42,43]. In order to explain the mechanism by which disorder is interpreted as a problem, Farrall et al. [1] incorporated several items into their model to measure these different forms of anxiety. To grasp social and political attitudes, the researchers included a series of questions on people's relationship to authoritarianism and another question on how much long-term community change was a concern to them. The respondents' perceptions of their local social environment were portrayed through their responses to several sets of questions about their concern regarding social cohesion, collective efficacy and changes in the behavior of young people from the neighborhood. Finally, the perception of disorder was measured using four different indicators. Overall, the model is based on a total of 29 questions on social and political attitudes and values alone. It remains the most comprehensive model to date for the study of the relationship between fear of crime and other forms of preoccupations.

The EEF model extends Ferraro's work downstream in a specific direction, namely in conceptualizing and operationalizing the emotional component of fear of crime. In the mid-1990s, the first questions used to measure this social phenomenon ["How safe do you feel walking alone in your neighborhood after dark?"] had already been the subject of strong criticism. Garofalo and Laub [66] and Garofalo [67] identified several problems with this, including the omission of the term 'crime', the vagueness of the term 'neighborhood' as a geographical space and placing respondents who do not go out at night in a perfectly hypothetical situation. A few years later, Ferraro and Lagrange [68] developed a set of proposals aimed at making the formulation of these questions clearer. The researchers suggested starting the questions with "how afraid…" to better evaluate the state of fear and ending them with "in your everyday life" to avoid hypothetical situations and, above all, by proposing to refer to specific types of harm. The latter proposal was inspired by the work of Warr and Stafford [69] and Warr [70] whose results showed the variation of levels of fear according to the types of harm that are feared and involved using several indicators to measure the emotional component of fear of crime. However the questions were still considered too imprecise. Firstly they did not measure any emotions other than fear [71,72] and also, more importantly, because they overestimated its intensity [73]. To attempt to solve this issue, Farrall and his colleagues stressed the importance of capturing tangible real-life experiences of fear [24,74]. The authors proposed adding three new questions in addition to the very general question used in the British Crime Survey ["How worried are you about being [burgled/robbed/having your car stolen/etc.]?" [1:pp.200]]. The first question asked whether respondents had been afraid of these different types of attacks in the previous 12 months, the second asked about the frequency of fears over the same period and the last asked about the emotional intensity of the last occurrence of such fear [75–77]. This increased measurement accuracy enabled Farrall et al. [1] to incorporate two distinct emotional responses into their model. The first was *anxiety* which they characterized as a person being 'quite' or 'very' worried despite having experienced no episode of fear in the previous 12 months. The second was *worry*, which was characterized by being 'quite' or 'very' worried while having experienced fear at least once in the previous 12 months. For the researchers the important issue was whether these two reactions are associated with the same factors and more broadly whether the underlying interpretive processes are identical in both cases.

- The results of the first version of the model

Farrall, Jackson and Gray decided to put the EEF model to a rigorous test by applying it to two different datasets. The first was the database deriving from the 2003/2004 British Crime Survey whose questionnaire exceptionally included the three new questions put forward by Farrall. The second was the results of a survey of public attitudes to crime carried out in a rural area in the north of England [4,78]. There was a dual advantage to using the British Crime Survey data. As well as the representativeness of its sample, the survey offered the possibility of matching external crime and deprivation data in different areas. However the British Crime Survey included very few questions on social and political attitudes. Conversely the results of the local survey could not be generalized outside the scope of the survey but presented the dual advantage of the large number of items included that related to social and political attitudes [THIS IS PARTICULARLY THE CASE OF THE BRITISH ELECTION SURVEY WHICH INCLUDES A LARGE NUMBER OF QUESTIONS ON OPINIONS. HOWEVER, THIS SURVEY DOES NOT MEASURE FEAR OF CRIME. THE NOVEL FEATURE OF JACKSON'S LOCAL SURVEY [4] IS THAT IT COMBINES QUESTIONS ON OPINIONS AND QUESTIONS AIMED AT MEASURING FEAR OF CRIME IN THE SAME QUESTIONNAIRE.] (the aforementioned 29 questions) and of focusing on a sample of 1 000 people from a same shared environment (which enabled researchers to focus on the processes of evaluating disorder).

When the EEF model was applied to the British Crime Survey data it showed that the level of crime and deprivation in a neighborhood has a moderate influence on the perception of disorder. In contrast, the subjective assessment of the environment strongly predicted concern about social cohesion and the perceived likelihood of victimization. Risk assessment in turn strongly influences the two emotional responses of anxiety and worry about crime. In this way, anxiety and worry are both associated with the perceived likelihood and perception of disorder but unlike the latter, the former is not predicted by victimization experiences.

When applied to local survey data, the EEF model showed that authoritarian attitudes and concern about the long-term decline of the community predict people's perceptions of disorder. Also, judging disorder to be a 'problem' influences concern about social cohesion and collective efficacy. These three cognitive evaluations influence the perceived likelihood of victimization which in turn predicts anxiety and worry about crime. To sum up, respondents who are more authoritarian regarding law and order, and more concerned about the long-term deterioration of the community, perceive more disorder in their immediate environment. This increases the likelihood that they will associate any such disorder with low social cohesion and low collective efficacy. Finally, these respondents are more likely than others to judge victimization to be a risk and to experience frequent and intense anxiety or worry.

These results led Farrall, Jackson and Gray to return to two concepts developed some years earlier by Jackson [4]. With a clearly stated objective of theorization, these researchers propose to consider *experiential fear* and *expressive fear*. The former is a response to a clearly identifiable *stimulus* like finding yourself in a dark street at night, hearing a strange noise behind you or being in the close vicinity of a group of young people. The latter refers to individuals' attitudes, representations and value judgments when expressing themselves on the subjects of crime and delinquency. Thus, fear of crime includes an *experiential dimension* – tangible fears experienced in a given situation – and an *expressive dimension* linked to people's attitudes towards social change, order, stability and cohesion [1]. The way these function in theoretical terms can be summarized as follows: *experiential fear* is influenced by a perception of disorder which in turn is predicted by *expressive fear* (the way a person perceives his or her community or neighborhood and also, more broadly, a person's relationship to society and its evolution). Working along these lines, Farrall, Jackson and Gray confirmed Ferraro's results (the perception of the environment influences the perceived likelihood of being a victim) and took them further to include the idea that political and social attitudes and values predict a person's perception of his or her environment.

- The results of the second version of the model

In an article published two years later, the same research team presents a new version of the EEF model tested on data from a survey conducted in 2007 with a representative sample of the London population [2]. Rather than analyzing the data using a structural equation model (as in the first version), the researchers establish a theoretical typology that first

distinguishes between the 'anxious' and the 'worried', and then between those who report that fear of crime reduces their quality of life (dysfunctional) and those who report that fear of crime does not reduce it (functional). The researchers identify five groups: the 'unworried', the 'anxious-functional', the 'worried-functional', the 'anxious-dysfunctional', and the 'worried-dysfunctional'. Using regression analysis, they then seek to identify the correlates of these different groups.

Concerning sociodemographic variables, gender is the only significant predictor. Being a woman increases the likelihood of belonging to the four groups of 'worried' and 'anxious'. Regarding the evaluation of the residential neighborhood, the perception of disorder is also a predictor for these same four groups. In contrast, collective efficacy is a contributing factor for membership only in the 'anxious-dysfunctional' and 'worried-dysfunctional' groups. Finally, regarding social and political attitudes and values, concern about moral decline [*In this article, the authors choose to discuss 'concern about moral decline' rather than 'authoritarian attitudes towards law and order.' However, the questions that make up these indicators (latent variables) are the same in both cases.*] is a predictor for the two 'worried' groups ('worried-functional' and 'worried-dysfunctional') but not for the 'anxious' groups. As for concern about long-term community decline, it is not a contributing factor for any class. Lastly, victimization increases the likelihood of belonging to the solely 'worried-dysfunctional' group [2].

The comparison of the results from the two versions of the EEF model leads to three observations. The first concerns the similarities. In both cases, the perception of disorder is an explanatory factor for anxiety about crime and worry about crime. The second observation relates to the potential contributions of the second version of the model. Specifically, the perception of collective efficacy would be a significant predictor only for the dysfunctional groups (and not for all 'anxious' and 'worried'); and victimization is a predictor for the sole 'worried-dysfunctional' group (and not for all 'worried'). Finally, the last observation concerns the variations. It pertains to the results for social and political attitudes and values. Here, it is not the weak statistical relationship between these attitude variables and the 'worried' and 'anxious' groups that is surprising. Indeed, the first version of the model establishes an indirect relationship between fear of crime emotions (worry and anxiety about crime) and political and social attitudes, linked through the perception of disorder and collective efficacy. The absence of a direct effect between these two categories of variables, once the model controls for the perception of disorder and collective efficacy, is therefore a consistent result. Conversely, the fact that concern about moral decline predicts membership in the two 'worried' groups is an unexpected finding. In light of this result, the most surprising aspect is that this social and political attitude does not predict membership in the 'anxious' groups, even though this emotion should be the one most directly associated with the *expressive* dimension ["*This dimension of fear of crime we call expressive because it is often not related to any specific event or episode in the respondent's life, but rather encapsulates a general sense of uneasiness on their part. In many cases, we feel this sense of uneasiness is not directly related to crime, but is an expression of wider concerns about the state of society today*" [1:*pp. 232*].].

This unexpected result raises two questions (which the first version of the model aimed to address). What is the exact role of neighborhood perception in the relationship between social/ political attitudes and fear of crime emotions (worry and anxiety about crime)? What is the precise nature of the relationship between social/ political attitudes and anxiety about crime on one side, and social/ political attitudes and worry about crime on the other?

• The results of the third version of the model

Due to a lack of additional studies, the second question currently remains unanswered. In contrast, the first question is central to the latest version of the EEF model conducted by Hirtenlehner and Farrall [3]. The response provided by these researchers is based on the confrontation of two paradigms. The first posits that fear of crime emotions is a component of a broader concern regarding globalization, and is thus closely associated with forms of anxiety related to economic and social changes. The second paradigm – adopted in the first version of the EEF model – suggests that fear of crime emotions and social and political attitudes interact indirectly through concerns about the state of the community.

Based on structural equation models (SEM), the results show that economic fears and social fears strongly predict fear of crime, and that incorporating the perception of disorder in the residential neighborhood as an intermediary variable only

slightly improves the model's explanatory power. Hirtenlehner and Farrall [3] conclude that fear of crime reflects broader concerns regarding societal change (first paradigm). Although this third version of the EEF model does not distinguish between worry and anxiety about crime, employs fewer variables, and introduces new items (economic fears and social fears), its results deserve to be compared with those of the two previous studies. They contrast with those of the first version by showing that emotions related to fear of crime are directly linked to global concerns about globalization. This represents a significant break that challenges Ferraro's [7] model based on the evaluation of the immediate environment. The results of Hirtenlehner and Farrall's [3] study also diverge from those of the second version of the EEF model. While Gray et al. [2] demonstrate a direct effect of concern about moral decline on worry about crime, they find no effect of this variable on anxiety about crime. Furthermore, concern about long-term community decline does not directly predict either worry or anxiety about crime. Thus, the direct effect of global concerns on fear of crime emotions is much more nuanced in the second version of the model than in the third. Table 1 below presents a summary of the results of the three versions of the EEF model.

## Proposed model and study design

In this article, we hypothesize that these discrepancies could be resolved by employing different data analysis methods. To understand this argument clearly, it is important to briefly discuss the analysis methods chosen by the developers of the EEF model.

As mentioned earlier, the first and third versions of the model are based on structural equation models (SEM). This method combines several aspects of statistical modeling, such as factor analysis and multiple regression, to analyze the structural relationships among multiple variables. It is a multivariate statistical technique used to test complex theoretical models. Indeed, the principle of the SEM is to use a hypothetical-deductive testing logic to determine *a priori* the structure of relations that are the best adapted to the data explaining the phenomenon being studied. These relations are said to be linear in that varying an input parameter proportionally (according to the value of the standardized coefficients) causes all others associated with it to be varied as well according to the structure of the relations determined by the model. This approach aims to bring out the 'own effect' of the independent variables determined *a priori* and tends to favor the reading of variations in terms of degree. *People who are more authoritarian regarding law and order are more likely to interpret disorder in their neighborhood as being threatening which increases the probability of them being worried or anxious at the idea of being a victim of a crime* [1].

This is why we think it is important to consider the existence of structures whose variations are of kind [27,28]. One of the characteristics of the EEF model is its use of a large number of variables. In such a situation, given the multitude

**Table 1. Comparative synthesis of the three versions of the EEF model.**

|  | Statistical Method | Type de structures | Main Results |
|---|---|---|---|
| Modèle EEF Version 1 | Structural Equation Model | A single structure of relationships between variables | . Social and political attitudes → neighborhood perceptions → worry and anxiety |
| Modèle EEF Version 2 | Theoretical typology + logistic regression analysis | Multiple structures of relationships between variables | . Neighborhood disorder perception → worry and anxiety<br>. Social and political attitudes → worry (but not anxiety) |
| Modèle EEF Version 3 | Structural Equation Model | A single structure of relationships between variables | . Social and political attitudes → worry and anxiety (measured indistinctly)<br>. Negligible effect of neighborhood disorder perception on worry/anxiety |

This table presents the statistical method, the type of relationship structure between the variables, and the main results of the three versions of the EEF model. The symbol "→" indicates an influence relationship and shows the direction of the relationship.

of possible parameters, adopting a hypothetico-deductive reasoning approach risks overlooking certain relationships between the variables. This risk is all the greater since SEMs are used to identify the most fundamental interaction structure, often at the expense of others. Thus, we hypothesize that in order to understand the nature of the discrepancies across the three versions of the EEF model, it is necessary to identify all existing combinations among the variables. Based on the results of the Ile-de-France victimization survey, Zauberman et al. [29] and Jardin et al. [26] show that worry about crime, concern about crime, perception of disorder and victimization are combined together in different ways in different areas of this region. This research used multivariate descriptive statistical analysis techniques and found there to be different structures of interaction between fear of crime's experiential and expressive components [Admittedly, SEMs are also capable of identifying such differences. The first version of the EEF model applied to data from the British Crime Survey shows, for example, that worry about crime can be triggered by having been the victim of a crime, a witness to a crime or by knowing someone who has been a victim of a crime in addition to the triggers described above. None of these variables is determined by moral or political values or even by a perception of disorder in a person's neighborhood. This clearly shows a difference in nature with the previous structure of interactions, and not just a difference in degree. However we do not consider the SEM to be the most suitable tool for identifying different configurations. Indeed, the addition of relations, variables or even conditions necessary to identify new combinations of variables could risk making the SEM very difficult to decipher (Moreover, in some cases the significance thresholds could be subjected to inferential constraints). This would deprive the model of a major part of its interest which derives from its capacity to reproduce the properties and fundamental mechanisms of fear of crime in a simplified way.].

It is precisely to move "beyond a dichotomy of worried/ not worried" [2:pp.86] that the second version of the model abandons structural equation modeling (SEM) in favor of typological analysis and regression analyses. This approach, as we have seen, allows for the identification of different groups of the 'worried' and 'anxious'. That being said, it does not address the critique we raised earlier. First, the typology used by Gray et al. [2] is based on only four variables. Certainly, regression analyses allow for the identification of the correlates of each class. But by excluding the relationships among the independent variables (several of which are components of expressive fear and experiential fear), this analysis loses the structural approach provided by SEM. For example, the relationships between social and political attitudes and the perception of the environment are completely excluded in the second version of the model. Therefore, it is impossible to determine whether these relationships vary according to each class. Next, the typology is theoretically constructed. The main risk of this practice is creating artificial classes that do not exist in social reality. In this sense, the very small size of some classes from this typology – 'worried-functional' comprising 5% of respondents and 'anxious-functional' 3% – indicate that these are at best extremely minority groups and at worst statistical artifacts.

Thus, all three versions of the EEF model employ statistical methods based on hypothetico-deductive reasoning. This observation should not be understood as a critique of these methods; rather, it suggests complementing them with other types of tools that rely on inductive reasoning. Given the large number of variables integrated into the EEF model, we favor *a posteriori* (inductive) reasoning over *a priori* (hypothetico-deductive) reasoning. In our view, *a priori* reasoning that starts from a principle to work back to the consequence has little chance of contributing to the discovery of new combinations of variables especially if these are little or poorly identified in the existing literature. To do this, we propose starting from *a posteriori* reasoning that works back from the consequences to the principles and therefore does not project *a priori* structures. Identifying the structures using this procedure should allow us to test three hypotheses:

H1: *There are relationship structures that associate the components of experiential fear with those of expressive fear*. According to this hypothesis, we should find the structure revealed by the first version of the EEF model.

H2: *There are also relationship structures that clearly separate the components of experiential fear from those of expressive fear*. Contrary to the results of the second version of the EEF model, we expect anxiety to be strongly associated with social and political attitudes.

H3: *Sociodemographic predictors vary significantly from one relationship structure to another*. We hypothesize that factors other than gender (as found in the second version of the EEF model) contribute to explaining and distinguishing the different relationship structures.

## Sample and data

Revisiting the model of experiential and expressive fear of crime requires a survey that includes variables rarely studied together. The survey must necessarily include questions about authoritarian attitudes towards law and order, attitudes towards social change in the community, questions related to the perception of disorder, concerns about social cohesion and collective efficacy. It must also incorporate questions about the experience of victimization, perceived likelihood, and be able to tell apart worry and anxiety about crime. In France – the country where the present study is conducted – the national victimization survey does not ask any questions about social and political attitudes and values, perceived likelihood of victimization, or social cohesion in the neighborhood. As for questions related to worry about crime, their wording do not take into account recent academic research developments. Thus, revisiting the EEF model was made possible by developing a new survey whose primary objective was a better understanding of the relationship between voting and attitudes towards crime. This survey, titled *Political Sociology of Insecurity during the 2022 Presidential Elections campaign* (SPIP2022), was carried out in the framework of the ELIPSS program.

The ELIPSS panel is an online survey instrument for the scientific community. It was set up in 2012 in the framework of the DIME-SHS EquipeEx [The Data, Infrastructures and Survey Methods in the Humanities and Social Sciences (ANR-10-EQPX-19–01) equipment of excellence (DIME-SHS EquipEx) is funded by the French government's Investments for the Future program and has been accredited by the University of Paris.] to study the evolution of behavior, situations and opinions in French society. The ELIPSS panel is coordinated by the Center for Socio-Political Data (CDSP) at Science-Po Paris and the National Center for Scientific Research (CNRS).

ELIPSS is based on a representative sample of the population of mainland France aged 18 and over in 2021. This sample was drawn in three phases. In 2012, an initial sample was randomly selected by the National Institute of Statistics and Economic Studies (INSEE) from the 2011 population census data. The individuals contacted were recruited through various methods: mail, phone, and face-to-face. With over 1 000 people joining the program, the recruitment rate for this operation was 27%. The first refresh occurred in 2016. Individuals selected from the 2014 population census were recruited face-to-face. More than 2 500 people then joined the program, with a recruitment rate of 32%. From September 2016 to September 2019, the panel sample ranged from 3 300–2 400 individuals. With an average annual attrition rate of 13%, around 400 individuals recruited in 2012 and 1 000 recruited in 2016 are still part of the ELIPSS panel today. In 2020, of the 2 400 individuals still participating, 1 400 agreed to continue their participation, to which 900 new panelists from the 2018 Fidéli database (reference database for tax data) were added, resulting in a recruitment rate of 14%. This refresh, initially planned as face-to-face, had to shift during the fieldwork to a multi-mode operation (excluding face-to-face) due to the Covid-19 pandemic. As of January 2021, the ELIPSS panel included 2 300 respondents.

The whole ELIPSS sample is invited to take part in a new survey every month, with an average fieldwork period of 4–5 weeks. The invitation is sent by email with a link to be clicked on. Respondents can thus complete the questionnaire using the digital device of their choice (computer, tablet, smartphone). A follow-up protocol by email and telephone is defined for each field. The average response rate to each survey is around 75%, with variations linked to the timing of the survey and the content of the questionnaire. It should be noted that panelists are encouraged to take part by regular and unconditional material rewards, mostly in the form of gift certificates. The optimal length a monthly questionnaire takes is 20 minutes and it never takes longer than 30 minutes.

Once a year, an Annual Survey is carried out on the sample population to collect precise information on the respondents' socio-demographic profile, family situation, housing situation along with a few major common social science indicators. These Annual Survey data enable a longitudinal follow-up of the sample and most variables (sometimes recoded) are systematically matched to other surveys carried out during the year in question.

Once a survey field has been closed, the data are post-produced (coding, anonymization checks, weighting calculations, etc.) and documented in accordance with FAIR principles by the CDSP then disseminated through the Quetelet-Progedo-Diffusion and data.sciencespo data repositories.

## Measures of variables

The SPIP2022 survey project was submitted in July 2021 and accepted by the ELIPSS Panel's Scientific and Technical Committee in September of the same year. Its aim was to study the relationship between crime (experienced and perceived) and political behavior during the 2022 French presidential election campaign. This gave us the opportunity to create a *sui generis* questionnaire combining many indicators taken from surveys in France and in English-speaking countries (see S1 Table in the Supporting information).

Authoritarian attitudes, concern about long-term community change, concern about social cohesion, perceived likelihood of victimization and worry about crime (frequency) were measured using the instruments developed by Farrall et al. [1].

Victimization, perception of neighborhood disorder and concern about collective efficacy are measured using instruments derived from the French victimization survey (Cadre de Vie et Sécurité, CVS) [THE FRENCH VICTIMIZATION SURVEY HAS RECENTLY BEEN COMPLETELY OVERHAULED. IN 2022, THE SURVEY WAS PUT UNDER THE RESPONSIBILITY OF THE FRENCH MINISTRY OF THE INTERIOR RATHER THAN THE INSTITUT NATIONAL DES STATISTIQUES ET DES ÉTUDES ÉCONOMIQUES, IT IS NOW CARRIED OUT IN TRI-MODE (AND NO LONGER ONLY FACE-TO-FACE) AND THE QUESTIONNAIRE HAS BEEN COMPLETELY RENEWED. ANOTHER AIM OF THE SPIP2022 SURVEY IS TO IDENTIFY THE EFFECTS OF THE ADMINISTRATION MODE AND QUESTIONNAIRES BY ASKING AN ONLINE PANEL SOME OF THE QUESTIONS FROM THE CVS SURVEY THAT WERE PREVIOUSLY CONDUCTED FACE-TO-FACE (SUBSEQUENTLY MODIFIED) FOR THE FIRST TIME. FOR THIS REASON, FRENCH QUESTIONS AND ISSUES WERE GIVEN PRIORITY TO MEASURE CERTAIN ASPECTS OF EXPERIENTIAL FEAR, EXPRESSIVE FEAR AND VICTIMIZATION.]. However, these questions were inspired by major victimization surveys in Britain and the United States. Therefore, they are very similar in their wording. Conversely, there is a notable difference between the SPIP2022 survey and the English surveys used by Farrall, Jackson, Gray, and Hirtenlehner. This stems from the introduction of an indicator that has been used for a long time in French victimization surveys that asks respondents to choose from a list the most concerning problem in French society. This question makes it possible to measure what Furstenberg [79] called *concern about crime* or, in other words, a value judgment which consists in placing crime at the forefront of the problems to be solved through governmental policies. Furstenberg considered this attitude should be dissociated from *fear of crime* which refers to the more tangible fear of being robbed or the victim of a physical assault on one's person or property. French research has clearly shown the specific features of these two dimensions. Concern about crime is particularly virulent among individuals with little education who are to the right of the political spectrum and find it extremely difficult to adapt to societal changes but *are not necessarily themselves confronted with the risks of victimization*. Alongside this, fear of crime refers to the perceived risk of victimization and experiences felt to have been threatening [26,29,80,81]. This finding closely corroborates one of Farrall, Jackson, and Gray's conclusions, namely when they stated "that emotional responses to risk and crime manifest in two ways: anxiety/*expressive fear*s (without moments of emotion that 'spike' in individual's daily lives, but rather bubble under in a more diffuse manner) and worry/*experiential fear*s (perhaps stimulated by more concrete cues of criminal threat)" [1:pp.227] [JACKSON HAD MADE THE SAME FINDING SOME YEARS EARLIER: "INTENSITY MEASURES MAY BRING OUT THIS EXPRESSIVE ASPECT OF THE PHENOMENON WITHOUT NECESSARILY AN EXPERIENTIAL ASPECT" [4:PP.961].]. While the SPIP2022 survey does not specifically measure anxiety, it does measure concern about crime along with a host of problems likely to be considered priorities by respondents. Furthermore, the SPIP2022 survey includes two questions aimed at measuring attitudes

towards crime. The first asks respondents about the causes of crime and the second what should be done to reduce the level of crime. Finally, since this survey does not include questions about the level of quality of life – which allow Gray et al. [2] to distinguish between *functional* and *dysfunctional* fear of crime – we will introduce a proxy variable for perceived happiness in the second analysis.

It should be noted that the SPIP2022 survey was administered to the ELIPSS panel between November 11th and December 16th 2021. The survey was completed by 1 648 panelists, representing a response rate of 75%. While the survey is based on a panel, this article focuses on a single survey campaign (cross-sectional data).

## Models and data analysis procedure

What methodological alternative can be adopted to attain a deeper understanding of the multidimensional structures of *experiential fear* and *expressive fear*? We believe that the most promising answer consists in carrying out a two-step statistical analysis, based on two complementary kinds of methods.

The first is a hierarchical clustering applied to multivariate response patterns. These statistical methods serves as a complement to modeling methods. Statistical modeling methods (of which SEM is one technique) are based on deductive reasoning. Their objective is to test hypotheses formulated about the expected relationship between variables. This deductive inference, by definition, requires applying *a priori* model to the data. In this case, observations are used to test conjectures. In contrast, multiple configurations analysis prioritizes observation over conjectures, following a different logic. These statistical methods seek to relax the constraints inherent in models by employing a technique that imposes no *a priori* structure on the data. The general logic involves starting from observation to "discover the dimensions of an existing equilibrium" [82:pp.48]. This is why this family of statistical methods is particularly well-suited for the purpose of reexamining the relationships between the variables of the EEF model to identify a range of different patterns. The active variables included in the analysis are those used in the first version of the EEF model, as they are all well-established in the literature on fear of crime [THE NEW VARIABLES INCLUDED IN THE SECOND AND THIRD VERSIONS OF THE MODEL (ECONOMIC FEAR, SOCIAL FEAR, PERCEIVED QUALITY OF LIFE RELATED TO FEAR OF CRIME) ARE MORE EXPLORATORY IN NATURE AND HAVE RARELY BEEN TESTED ELSEWHERE. IN THE CONTEXT OF THE ANALYSIS OF MULTIPLE CONFIGURATIONS, WE THEREFORE CHOOSE TO RETAIN ONLY THE VARIABLES THAT ARE MOST CLEARLY IDENTIFIED IN THE SCIENTIFIC LITERATURE. HOWEVER, THE REGRESSION ANALYSIS WILL INCLUDE A PROXY VARIABLE FOR PERCEIVED QUALITY OF LIFE.]. As a reminder, the variables involved in the analysis are worry about crime, perceived likelihood, victimization, authoritarian attitudes, attitude towards crime, concern about change in the community over the long term, concern for social cohesion, perception of disorders in the neighborhood, concern for collective efficacy and priority problem in French society. For this first step, the analysis process is organized as follows. The dataframe organized with rows (individual respondents) and columns (variables) is processed through a multiple component analysis. Geometrically, this table corresponds to a cloud of 1 648 points (number of individual respondents) located in a high-dimensional space (number of variables selected for the analysis). MCA allows us to identify the major synthetic dimensions of this multidimensional space and to visually represent them in two dimensions, in order to highlight the strongest statistical relationships that exist among the variables under consideration [*GEOMETRIC METHODS, OF WHICH MULTIPLE COMPONENT ANALYSIS (MCA) IS A SPECIFIC CASE, ALLOW FOR THE CONSTRUCTION OF A SOCIAL SPACE, THAT IS, DEFINING A DISTANCE BETWEEN STATISTICAL INDIVIDUALS BASED ON THE SELECTED VARIABLES (REFERRED TO AS ACTIVE VARIABLES) FOR THIS PURPOSE. INDIVIDUALS ARE THEN REPRESENTED AS A CLOUD OF POINTS IN A MULTIDIMENSIONAL SPACE. ONCE THE SPACE IS DEFINED BY THE CHOICE OF ACTIVE VARIABLES, GEOMETRIC DATA ANALYSIS INVOLVES REDUCING THE NUMBER OF DIMENSIONS OF THIS SPACE BY CREATING A NEW SYSTEM OF AXES (REFERRED TO AS PRINCIPAL DIMENSIONS, FACTORIAL AXES, ETC.). THIS NEW SYSTEM OF AXES IS SUCH THAT THE DISPERSION (INERTIA) OF THE CLOUD PROJECTED ONTO THE FIRST DIMENSION IS MAXIMUM (MEANING THAT, ON THIS AXIS, THE INERTIA OF THE CLOUD IS AS HIGH AS POSSIBLE), AND SO ON FOR THE SUBSEQUENT DIMENSIONS*" [83:pp.80].]. Then, we extract from the output of the MCA a specific data structure called a complete disjunctive table. This table is organized with rows (individuals respondents) and columns (modalities of each variable). As a consequence, this stage of the analysis transforms the dataframe, replacing variable columns with modalities columns. The use of the complete disjunctive table allows for the separate

treatment of each variable's modality and the filtering of the most relevant modalities without excluding non-responding individuals from the processed data. This reduces sample distortion and helps to preserve its representative-ness. The approach followed corresponds to the specific MCA method [84]. The dataframe is then filtered, as modalities coding for missing values are removed. Finally, the constructed dataframe is analyzed through a hierarchical clustering based on euclidean metric and ward distances. Classification allows for a qualitative synthesis of the results of MCA. The objective is to group individuals into a small number of homogeneous classes, but clearly different from each other. Thus, the diversity of a sample is reduced to a few contrasting classes. In other words, this technique aims to summarize a large amount of information in order to highlight the existing prominent features in the data through the development of typol-ogies. Here again, the grouping procedure operates inductively. In the case of hierarchical clustering, there are no initial hypotheses about the data structure or the number of clusters (for example, the clusters are not built based on presup-posed taxonomic criteria). The method follows the observed data and constructs groupings by minimizing intra-group variance (see the formula below and the introduction to the results section for a description of how hierarchical clustering work). We use the state of art algorithm by Daniel Müllner [85], with A and B two observations separated by distance d. The formula estimates the distances between the appoint according to the ward distance criteria and agglomerate literately sets of points (denoted $x$) that minimize such distance up to a threshold defined ex post given the interpretation of the dendro-gram. μ is the mean value of observed for A and B.

$$\frac{|A| \cdot |B|}{|A \cup B|} \left\| \mu A - \mu B \right\|^2 = \sum_{x \in A \cup B} \left\| x - \mu A \cup B \right\|^2 - \sum_{x \in A} \left\| x - \mu A \right\|^2 - \sum_{x \in B} \left\| x - \mu B \right\|^2$$

The second step is the computation of a logistic regression. Based on traditional sociodemographic models, the regression analysis aims to determine the respective weight of these main factors on the different classes, resulting from the previous analysis. The use of regression allows for the reintegration of an inferential statistics approach as an extension of struc-tural description. By measuring the influence of one variable on another "holding all else constant," which means taking into account other independent variables introduced into the model, regression analysis seeks to determine the own effect of each independent variable. In other words, this method allows, unlike MCA and classification, to untangle the maze of structural effects by identifying potential confounding variables (in statistical modeling methods, variables that simultane-ously influence the independent and dependent variables). Therefore, the value of testing a sociodemographic model using regression lies in the ability of this method to measure the adjusted effect of each sociodemographic factor (e.g., gender) on different classes while controlling for the potential effect of other variables integrated into the model (age, education level, region of residence, etc.). The ultimate goal is to retest one of the most commonly used models in the study of fear of crime – the sociodemographic model [21,30] – to strengthen the statistical reliability of the results from the MCA and classification by identifying the social correlates of each of the four classes. Technically, cluster membership for each class is placed as the dependent variable and a set of sociodemographic predictors as independent variables. The number of classes consequently determines the number of categories of the dependent variable in the performed regression. The independent variables included in the model are sex, age, level of education, perceived standard of living, type of housing, region of residence, and perceived level of happiness. For each outcome we specify the following model equation:

$$Ln\left(\frac{p\,(X|1)}{p\,(X|0)}\right) = a0 + a1_{sex} + a2_{age} + a3_{education} + a4_{standard\ of\ living} + a5_{dwelling} + a6_{region} + a7_{happiness}$$

where;
   $Ln\,(p(X|1)/p(X|0))$ express the conditional probability of the outcome coded 1 as opposed to any other outcome;
   $a0$ is the constant of the model;

$a1, a2… a7$ are the regression coefficients for each independent variable [IN THIS SPECIFIC CONTEXT, WE'VE SELECTED THE LOGISTIC REGRESSION AS THE MOST CONVENIENT METHOD TO DEAL WITH A SET OF BINARY DEPENDENT VARIABLES. OUR MAIN INTEREST HERE IS NOT TO STUDY THE INNER WORKINGS OF THE RELATION BETWEEN EACH SOCIODEMOGRAPHIC VARIABLE AND OUR CLUSTERS OF ATTITUDES TOWARD CRIME. THE MAIN GOAL OF THIS SET OF LOGISTIC REGRESSIONS IS TO TEST THE STATISTICAL SIGNIFICANCE OF THE SOCIODEMOGRAPHIC SPECIFICITY OF EACH CLUSTER WHEN COMPARED TO THE REMAINING SAMPLE. AS THE CLUSTERING METHOD EMPLOYED DOESN'T, IN ITSELF, INCLUDE ANY INFERENTIAL INDICATOR, WE LEVERAGE THE LOGISTIC REGRESSION TO PROVIDE SUCH TESTS TO STRENGTHEN OUR RESULTS (T-TEST AND $R^2$)].

In summary, this two-step analysis aims to achieve a dual objective. The multivariate pattern analysis aims to identify the structures of relationships present in the data from an *a posteriori* reasoning that works back from the consequences to the principles (and not the reverse). The regression analysis then employs inferential reasoning to determine the sociodemographic correlates for each class.

## Results

Fig 1 presents the dendrogram (or hierarchical tree) from the hierarchical cluster analysis (HCA). The dendrogram is a tree-like diagram that illustrates the hierarchy of groupings among individuals in the context of hierarchical cluster analysis (HCA). Each branch of the dendrogram represents the progressive merging of clusters based on their similarity (distance), with the height reflecting the level of dissimilarity. It allows for the visualization of the data structure and helps determine the optimal number of groups by cutting the tree at a certain level.

The base is made up of all the individuals (1 cluster = 1 individual). At each step, two nearby clusters are merged into a new cluster. Ward's method determines which clusters to merge by minimizing the increase in intra-group variance. Thus, the calculation of the distance between each of the clusters makes it possible to iteratively increase the number of individuals per cluster while reducing the number of clusters. The partition (number of clusters selected) therefore depends on

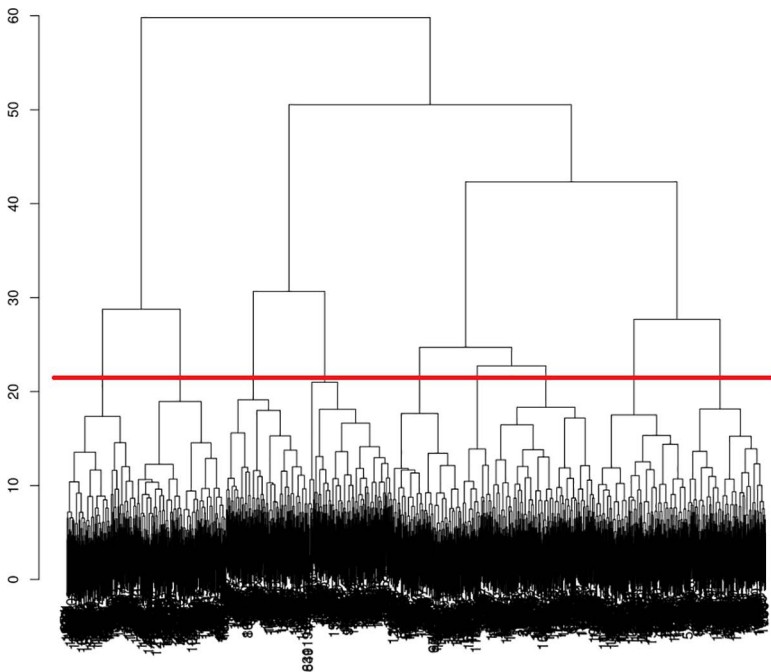

**Fig 1. Cluster dendrogram.**

the cut level on the hierarchical tree (dendrogram). The higher the division, the lower the number of clusters retained, but the greater the infra-class differences.

To extract highly detailed information from this clustering scheme, we select the first 9 clusters from the original dendrogram. This slice level (indicated by the horizontal red line in the dendrogram) has the advantage of capturing several structures of relationship between variables. On the other hand, it has the major disadvantage of having to describe too many classes for the format of an academic article. Moreover, our goal is not to present all existing configurations, but to identify the 'fundamental' relational structures. In other words, we seek structures that, in relation to others, either exhibit variations in degree that reflect strong opposing relationships, or variations in nature that present original kinds of relationships. Thus, we have gathered together 1) clusters with small variations in degree (two classes with the same structure of relation, but with slightly different levels of intensity) and 2) clusters whose discriminating variations are based on distinct societal concerns, without introducing differences in the nature of the relationship to experiential and expressive fear of crime (two groups with the same structure of relation, but the former declares to be primarily concerned with health and the latter with poverty issues). This operation, using a dynamic threshold to cut the dendrogram [86] provides a condensed clustering scheme containing 4 main clusters [Aɴxɪous' (ᴄʟᴜsᴛᴇʀ 1 + ᴄʟᴜsᴛᴇʀ 2); 'ᴜɴᴡᴏʀʀɪᴇᴅ' (ᴄʟᴜsᴛᴇʀ 3 + ᴄʟᴜsᴛᴇʀ 4 + ᴄʟᴜsᴛᴇʀ 6 + ᴄʟᴜsᴛᴇʀ 8 + ᴄʟᴜsᴛᴇʀ 9); 'ᴡᴏʀʀɪᴇᴅ-ᴅʏsꜰᴜɴᴄᴛɪᴏɴᴀʟ' (ᴄʟᴜsᴛᴇʀ 5); 'ᴡᴏʀʀɪᴇᴅ-ꜰᴜɴᴄᴛɪᴏɴᴀʟ' (ᴄʟᴜsᴛᴇʀ 7).].

### Results of the cluster analysis

To identify and manipulate these classes more easily, we chose to use the labels employed by Gray et al. [2]. However, it is important to be particularly cautious. Using these labels does not mean that the classes in the two studies are entirely comparable. There are sometimes significant differences between them, which will be discussed in discussion section. Nevertheless, to avoid excessively multiplying names and concepts, it seems preferable to frame our results within the terminology used by the designers of the EEF model whenever possible.

- The 'unworried'

Half of the respondents (50.2%) were in this group. First and foremost it is characterized by very low exposure to any risk of victimization as is shown by the over-representation in this group of respondents who say they were not a victim in the past 12 months. The same is the case for the worry about crime and perceived likelihood. A higher proportion of the 'unworried' than from the French population as a whole *have never been worried about crime* and *consider it very unlikely* they could be burgled, robbed or assaulted (Table 2).

This group is also characterized by a very positive assessment of the environment they live in. Respondents who consider crime, youth gangs, family violence, drug sales and use or bad reputation to be a *problem* in their neighborhood are underrepresented compared to the average. Social cohesion is also a source of a high level of satisfaction. This group features the highest proportion of respondents who say they live in a *friendly, tight-knit community* in which the *neighbors are trustworthy* and *look after each other*. The 'unworried' are also more likely to rate collective efficacy in a positive light. More than anywhere else, these respondents believe they *can rely on their neighbors to* make a purchase, babysit a child for a few hours, get around, call the emergency services if needed or notify the police if necessary. The perception of long-term social change in the community reflects how long-term this satisfaction is. More 'unworried' people than the average consider the quality of their neighborhood relations to be *stable* (Table 3).

In terms of social and political attitudes, this class is defined by a low concern for safety issues. Less than elsewhere, *crime* is designated as the priority problem in French society. On the other hand, *environmental* and *poverty* concerns are at their highest level here. Moreover, it is among the 'unworried' that we find the highest proportion of respondents incriminating poverty to explain crime. Asked about the best solutions to fight against this phenomenon, these respondents said they are more favorable than the others to the *development of youth prevention* and more unfavorable to *building new*

**Table 2. Perceived likelihood, victimization and worry about crime (unworried).**

| Latent variable | Manifest variable | Modalities | Class (%) | Sample (%) | Odd Ratio |
|---|---|---|---|---|---|
| Perceived likelyhood of being victimized | Physical assault | Very unlikely | 57 | 49 | 1,4 |
| | Sexual assault | Very unlikely | 74 | 68 | 1,3 |
| | Theft of a vehicle | Very unlikely | 52 | 46 | 1,3 |
| | Larceny | Very unlikely | 52 | 45 | 1,3 |
| | Verbal assault | Very unlikely | 42 | 32 | 1,6 |
| Experienced victimization | Bike theft | Never | 98 | 95 | 2,6 |
| | Burglary | Never | 97 | 94 | 2,0 |
| | Car theft | Never | 99 | 97 | 5,0 |
| | Damage of the housing | Never | 98 | 95 | 3,7 |
| | Damage of vehicle | Never | 93 | 85 | 2,2 |
| | Insults | Never | 84 | 72 | 2,1 |
| | Motorbike theft | Never | 100 | 98 | 8,7 |
| | Physical assault | Never | 99 | 94 | 5,4 |
| | Sexual assault | Never | 100 | 98 | DIV/0 |
| | Threats | Never | 93 | 83 | 2,6 |
| Worry about crime | Burglary | Never | 82 | 74 | 1,7 |
| | Physical assault | Never | 95 | 86 | 3,2 |
| | Sexual offense | Never | 97 | 92 | 2,4 |
| | Larceny | Never | 93 | 84 | 2,5 |
| | Vehicle theft | Never | 91 | 82 | 2,2 |
| | Verbal assault | Never | 86 | 70 | 2,6 |

The column 'class' corresponds to the proportion of respondents concerned from the group (%), the column 'sample' to the proportion of respondents concerned in the general population (%), and the 'odds ratio' column represents the value of the odds ratio relative to the difference between these two proportions.

---

*prisons*. Finally, the 'unworried' are characterized by pronounced attitudes against authoritarian positions. Respondents who do not *agree at all* or *rather disagree* that schools should teach children to obey authority and that young people today do not have enough respect for traditional French values are over-represented in this group compared to the average. The same is the case for respondents who disagree (*rather disagree*) with the idea of giving stiffer sentences to people who break the law (Table 4).

To sum up, the 'unworried' are characterized by a very low exposure to risk, a very low perceived likelihood of victimization and almost non-existent worries about crime. The neighborhood and relations with residents are viewed positively. Health and the environment are seen as priority concerns rather than crime. Authoritarian attitudes are not very widespread and even more or less firmly rejected.

• The 'worried-dysfunctional'

This group brings together just over than a tenth of the whole survey population (11.7%). It is presented as the perfect countertype to the previous case. The 'worried-dysfunctional' have a high level of exposure to the risk of victimization. More than the average, members of this group report having been burgled, robbed or assaulted in the previous 12 months, sometimes more than once. There was also a more widespread perceived likelihood of being a victim. Respondents who considered the chances of being victimized to be *quite likely* or even *very likely* are clearly over-represented compared to the average. The same is the case with worries about crime which are more frequent and more diverse here than in the national average (Table 5).

**Table 3. Perception of local environment (unworried).**

| Latent variable | Manifest variable | Modalities | Class (%) | Sample (%) | Odd Ratio |
|---|---|---|---|---|---|
| The attitudes toward social change in the community | A sense of belonging to the community | Not changed | 68 | 61 | 1,4 |
| | A sense of right and wrong amongst people who live here | Not changed | 75 | 60 | 2,0 |
| | A sense of shared value amongst people who live here | Not changed | 71 | 60 | 1,7 |
| | Young people's respect for other | Not changed | 69 | 50 | 2,2 |
| | Young people's respect for rules and authority | Not changed | 69 | 49 | 2,3 |
| The concern about collective efficacy | I can rely on my neighbors to call the emergency services if needed | Yes | 87 | 81 | 1,7 |
| | I can rely on my neighbors to notify the police if necessary | Yes | 75 | 68 | 1,4 |
| | I can rely on my neighbors to make a purchase | Yes | 62 | 56 | 1,3 |
| | I can rely on my neighbors to babysit a child for a few hours | Yes | 39 | 31 | 1,4 |
| | I can rely on my neighbors to help me get around | Yes | 45 | 39 | 1,3 |
| The concern about social cohesion | People in this area are trutworthy | Yes | 39 | 30 | 1,5 |
| | This area is a friendly place to live | Yes | 79 | 70 | 1,6 |
| | This area is a place where local people look after each other | Yes | 36 | 31 | 1,3 |
| | This area has a close, tight-knit community | Yes | 11 | 8 | 1,5 |
| The perception of problems in neighborhood | Bad reputation | Yes | 3 | 8 | 0,4 |
| | Crime | Yes | 3 | 18 | 0,2 |
| | Drug | Yes | 10 | 22 | 0,4 |
| | Family violence | Yes | 5 | 10 | 0,4 |
| | Youth gangs | Yes | 4 | 13 | 0,3 |

The column 'class' corresponds to the proportion of respondents concerned from the group (%), the column 'sample' to the proportion of respondents concerned in the general population (%), and the 'odds ratio' column represents the value of the odds ratio relative to the difference between these two proportions.

This group is also characterized by a very pronounced dissatisfaction of the respondents with their living environment. The proportion of those who consider crime, vandalism and nuisance to be a *problem* in their neighborhood reached a peak. Moreover, this group feature the lowest proportion of respondents who say they live in a *friendly* neighborhood. Less than elsewhere, the 'worried-dysfunctional' say they are satisfied with neighborhood relations: those who feel they could *rely on their neighbors* for help in their daily lives are underrepresented in this group. This level of pronounced dissatisfaction with their neighborhood is accompanied by a strong concern about a long-term deterioration of the community. There is an over-representation in the 'worried-dysfunctional' group of respondents who consider neighborhood life, the morality of other residents and the feeling of sharing the same values as other members of the community to be *declining*. This representation of a declining community is even more pronounced in this group as regards young people and their behavior (Table 6).

The 'worried-dysfunctional' also differ from the 'unworried' by clearly identifying *crime* as a priority problem in French society. These respondents were also more likely to display authoritarian attitudes than the average. There is an over-representation in the 'worried-dysfunctional' group of respondents who *fully agree* that people who break the law should

**Table 4. Social and political attitudes (unworried).**

| Latent variable | Manifest variable | Modalities | Class (%) | Sample (%) | Odd Ratio |
|---|---|---|---|---|---|
| If we could spend more money on fighting crime | Build new prisons | Yes | 7 | 11 | 0,6 |
| | Develop youth prevention | Yes | 77 | 68 | 1,6 |
| Priority problem in French society | Poverty | Yes | 24 | 19 | 1,3 |
| | Environment | Yes | 25 | 18 | 1,5 |
| | Crime | Yes | 9 | 15 | 0,6 |
| The authoritarian and attitudes toward law and order | People who break the law should be given stiffer sentences | Rather disagree | 20 | 15 | 1,4 |
| | Schools should teach children to obey authority | Not agree at all | 12 | 9 | 1,4 |
| | Schools should teach children to obey authority | Rather disagree | 37 | 32 | 1,3 |
| | Young people today don't have enough respect for traditional French values | Not agree at all | 10 | 8 | 1,3 |
| | Young people today don't have enough respect for traditional French values | Rather disagree | 45 | 34 | 1,6 |
| The main cause of crime | Poverty | Yes | 27 | 22 | 1,3 |

The column 'class' corresponds to the proportion of respondents concerned from the group (%), the column 'sample' to the proportion of respondents concerned in the general population (%), and the 'odds ratio' column represents the value of the odds ratio relative to the difference between these two proportions.

be punished more severely, that young people today do not have enough respect for traditional French values and that schools should teach authority. Asked about the main causes of crime, these respondents are more numerous than the average to incriminate the *lack of judicial sanction*. In return, they quite categorically reject *poverty* as the main factor of crime. Regarding support for solutions, *building new prisons* and *recruiting more police officers* both obtain higher approval rates than in other groups. On the other hand, the solution of *preventive measures to avoid juvenile crime* is much less popular with this group than with others (Table 7).

In short, the 'worried-dysfunctional' are extremely archetypical. They are over-exposed to the risk of victimization, more often afraid of being victimized and are very dissatisfied with their living surroundings or neighborhood social relations. They also show very strong concerns about crime associated with pronounced authoritarian attitudes.

• The 'anxious'

This group comprises a significant quarter of the respondents (26.0%). Like the 'unworried', 'anxious' are defined by their marked under-exposure to robbery and assault, very infrequent worries about crime and a low perceived likelihood of victimization (Table 8).

However, this quasi-absence of experienced and perceived crime in daily life in the 'anxious' is not accompanied, as for the 'unworried', by a very high level of satisfaction regarding the local environment. 'Anxious' perceptions with regard to collective efficacy, social cohesion and disorders do not differ from those of the national average. There is only a slight over-representation in this group of respondents who point out the *lack of road safety* in their neighborhood.

The perception of social change in the community marks the first real break with the opinions of the 'unworried' group. For the 'anxious', "rather" high level of satisfaction with local environment (whose level remains much higher than that of the 'worried-dysfunctional" group) is not associated with a "rather" low level of concern about social change in the neighborhood. Admittedly there is an over-representation in this group of respondents who consider the sense of belonging to the community to be *stable* but more respondents than the average think young people's respect for others and for authority *has decreased* (Table 9).

**Table 5. Perceived likelyhood, victimization and worry about crime (worried-dysfunctional).**

| Latent variable | Manifest variable | Modalities | Class (%) | Sample (%) | Odd Ratio |
|---|---|---|---|---|---|
| Perceived likelyhood of being victimized | Burglary | Quite likely | 20 | 12 | 1,8 |
| | Physical assault | Quite likely | 26 | 8 | 4,1 |
| | Larceny | Quite likely | 32 | 11 | 3,7 |
| | Vehicle theft | Quite likely | 21 | 9 | 2,5 |
| | Verbal assault | Quite likely | 46 | 19 | 3,6 |
| | Verbal assault | Very likely | 13 | 3 | 4,1 |
| Experienced victimization | Burglary | Once | 7 | 4 | 1,7 |
| | Damage of the housing | Once | 10 | 3 | 3,2 |
| | Damage of vehicle | Once | 25 | 12 | 2,4 |
| | Insults | Once | 37 | 20 | 2,4 |
| | Insults | More than once | 20 | 7 | 3,3 |
| | Physical assault | Once | 15 | 4 | 4,2 |
| | Threats | Once | 30 | 12 | 3,2 |
| | Threats | More than once | 13 | 4 | 4,0 |
| Worry about crime | Burglary | Once | 16 | 13 | 1,3 |
| | Burglary | More than once | 20 | 12 | 1,9 |
| | Physical assault | Once | 23 | 8 | 3,6 |
| | Physical assault | More than once | 10 | 5 | 2,3 |
| | Larceny | Once | 14 | 7 | 2,2 |
| | Larceny | More than once | 17 | 8 | 2,4 |
| | Vehicle theft | Once | 19 | 9 | 2,4 |
| | Vehicle theft | More than once | 18 | 9 | 2,4 |
| | Verbal assault | Once | 35 | 18 | 2,5 |
| | Verbal assault | More than once | 30 | 10 | 3,6 |

The column 'class' corresponds to the proportion of respondents concerned from the group (%), the column 'sample' to the proportion of respondents concerned in the general population (%), and the 'odds ratio' column represents the value of the odds ratio relative to the difference between these two proportions.

This divergence is confirmed regarding moral and political values. The 'anxious' give an important place to *crime* and *terrorism* as priority problems in French society, especially with regard to their low exposure to the risk of victimization. Respondents from this group also show a strong propensity for authoritarian attitudes. There is an over-representation of respondents in this group who *fully agree* that schools should teach children to obey authority, that people who break the law should be given stiffer sentences and that young people today do not have enough respect for traditional French values. Those who *rather agree* with the latter assertion are also over-represented here. When asked about the main causes of crime, these respondents always tend to blame institutions (the *family*) rather than social factors and contexts (*poverty*). As for recommended solutions, the repressive proposals which consist in *building new prisons* and *recruiting more police officers* are more prevalent in this group than the national average (Table 10).

To sum up, the 'anxious' can be defined by their very low level of exposure to the risk of victimization, a low perceived likelihood and very infrequent worry of such an occurrence. But, at the same time, these respondents are highly concerned about crime. They also express significant concerns about social changes in the community (especially concerning the behavior of young people), coupled with pronounced authoritarian and security-oriented attitudes.

• The 'worried-functional'

**Table 6. Perception of local environment (worried-dysfunctional).**

| Latent variable | Manifest variable | Modalities | Class (%) | Sample (%) | Odd Ratio |
|---|---|---|---|---|---|
| The attitudes toward social change in the community | A sense of belonging to the community | Had decreased | 59 | 24 | 4,1 |
| | A sense of right and wrong amongst people who live here | Had decreased | 82 | 31 | 9,8 |
| | A sense of shared value amongst people who live here | Had decreased | 75 | 29 | 7,4 |
| | Young people's respect for other | Had decreased | 88 | 43 | 9,2 |
| | Young people's respect for rules and authority | Had decreased | 88 | 45 | 8,9 |
| The concern about collective efficacy | I can rely on my neighbors to call the emergency services if needed | Yes | 58 | 81 | 0,3 |
| | I can rely on my neighbors to help me get around | Yes | 21 | 39 | 0,4 |
| | I can rely on my neighbors to make a purchase | Yes | 37 | 56 | 0,5 |
| | I can rely on my neighbors to notify the police if necessary | Yes | 48 | 68 | 0,4 |
| The concern about social cohesion | This area is a friendly place to live | Yes | 24 | 70 | 0,1 |
| | This area is a place where local people look after each other | Yes | 13 | 31 | 0,3 |
| | I am proud to live in this neighboorhood | Yes | 11 | 27 | 0,3 |
| | People in this area are trutworthy | Yes | 14 | 30 | 0,4 |
| The perception of problems in neighborhood | Bad reputation | Yes | 27 | 8 | 4,5 |
| | Crime | Yes | 44 | 18 | 3,6 |
| | Drug | Yes | 61 | 22 | 5,7 |
| | Family violence | Yes | 27 | 10 | 3,3 |
| | Youth gangs | Yes | 51 | 13 | 6,9 |

The column 'class' corresponds to the proportion of respondents concerned from the group (%), the column 'sample' to the proportion of respondents concerned in the general population (%), and the 'odds ratio' column represents the value of the odds ratio relative to the difference between these two proportions.

This group representing a eighth of the population (12.1%). Like the 'worried-dysfunctional', the 'worried-functional' are at a higher-than-average risk of victimization. There is an over-representation of respondents who reported having suffered a *burglary* or an *assault* in the past 12 months. Even more than the 'worried-dysfunctional', the 'worried-functional' are over-exposed to repeated crimes, particularly verbal assaults. Moreover, the worry about crime is also more prevalent among these respondents: there is an over-representation of 'worried-functional' who reported being afraid of being burgled, verbally assaulted, having their car stolen, sexually assaulted, robbed without violence or physically assaulted *once* in the past 12 months. Similarly, repeated experiences of worry are more common in this group than any other.

The perceived likelihood of being a victim of an assault or larceny is presented in a particularly interesting form in this group. Quite consistently, a relatively large proportion of respondents from this group consider the risk of being insulted/threatened, assaulted, burgled and robbed without violence in the coming year as *quite likely*. However, more than average, the 'worried-functional' consider it *quite unlikely* they might be sexually or physically assaulted which may reflect the relatively infrequent occurrence of the most violent assaults even within this group (Table 11). It could also be due to the 'worried-functional' having a good knowledge of the real risks of victimization. Based on experience, the most serious attacks could be considered rare, but this does not stop people from being afraid of such assaults from time to time, particularly in the presence of signal disorders.

**Table 7. Social and political attitudes (worried-dysfunctional).**

| Latent variable | Manifest variable | Modalities | Class (%) | Sample (%) | Odd Ratio |
|---|---|---|---|---|---|
| Priority problem in French society | Crime | Yes | 32 | 15 | 2,7 |
| If we could spend more money on fighting crime | Build new prisons | Yes | 20 | 11 | 2,1 |
| | Develop youth prevention | Yes | 41 | 68 | 0,3 |
| | Recruit more police officers | Yes | 29 | 17 | 2,0 |
| The authoritarian and attitudes toward law and order | People who break the law should be given stiffer sentences | Fully agree | 50 | 25 | 3,0 |
| | Schools should teach children to obey authority | Fully agree | 29 | 15 | 2,3 |
| | Young people today don't have enough respect for traditional French values | Fully agree | 35 | 14 | 3,4 |
| The main cause of crime | Poverty | Yes | 13 | 22 | 0,5 |
| | The lack of justice sanction | Yes | 32 | 17 | 2,2 |

The column 'class' corresponds to the proportion of respondents concerned from the group (%), the column 'sample' to the proportion of respondents concerned in the general population (%), and the 'odds ratio' column represents the value of the odds ratio relative to the difference between these two proportions.

**Table 8. Perceived likelihood, victimization and worry about crime (anxious).**

| Latent variable | Manifest variable | Modalities | Class (%) | Sample (%) | Odd Ratio |
|---|---|---|---|---|---|
| Perceived likelyhood of being victimized | Physical assault | Very unlikely | 54 | 49 | 1,3 |
| | Sexual assault | Very unlikely | 73 | 68 | 1,3 |
| | Larceny | Very unlikely | 52 | 45 | 1,3 |
| | Verbal assault | Unlikely | 49 | 43 | 1,3 |
| Experienced victimization | Burglary | Never | 96 | 94 | 1,7 |
| | Damage of the housing | Never | 97 | 95 | 2,2 |
| | Damage of vehicle | Never | 87 | 85 | 1,3 |
| | Insults | Never | 82 | 72 | 1,7 |
| | Motorbike theft | Never | 99 | 98 | 3,0 |
| | Physical assault | Never | 98 | 94 | 3,1 |
| | Sexual assault | Never | 100 | 98 | 3,7 |
| | Threats | Never | 91 | 83 | 2,0 |
| Worry about crime | Burglary | Never | 81 | 74 | 1,5 |
| | Physical assault | Never | 92 | 86 | 2,0 |
| | Sexual offense | Never | 96 | 92 | 1,9 |
| | Theft of a vehicle | Never | 87 | 82 | 1,5 |
| | Larceny | Never | 92 | 84 | 2,1 |
| | Verbal assault | Never | 77 | 70 | 1,4 |

Note : the column 'class' corresponds to the proportion of respondents concerned from the group (%), the column 'sample' to the proportion of respondents concerned in the general population (%), and the 'odds ratio' column represents the value of the odds ratio relative to the difference between these two proportions.

Indeed, this experiential fear – more widespread than in the rest of the sample – is associated with a perception of a greater number of problems in the neighborhood. This result contributes to explaining why fewer 'worried-functional' said they are living in the *tight-knit community* where residents are *trustworthy*. However, their awareness of living in an area where the risk of victimization is not negligible is not accompanied with systematically declinist attitudes regarding the community and neighborhood social relations, as is the case for the 'worried-dysfunctional'. Admittedly, 'worried-functional'

**Table 9. Perception of local environment (anxious).**

| Latent variable | Manifest variable | Modalities | Class (%) | Sample (%) | Odd Ratio |
|---|---|---|---|---|---|
| The attitudes toward social change in the community | A sense of belonging to the community | Not changed | 66 | 61 | 1,3 |
| | Young people's respect for rules and authority | Had decreased | 63 | 45 | 2,1 |
| | Young people's respect for other people | Had decreased | 58 | 43 | 1,8 |
| The perception of problems in neighborhood | Lack of road safety | Yes | 27 | 20 | 1,5 |

The column 'class' corresponds to the proportion of respondents concerned from the group (%), the column 'sample' to the proportion of respondents concerned in the general population (%), and the 'odds ratio' column represents the value of the odds ratio relative to the difference between these two proportions.

**Table 10. Social and political attitudes (anxious).**

| Latent variable | Manifest variable | Modalities | Class (%) | Sample (%) | Odd Ratio |
|---|---|---|---|---|---|
| Priority problem in French society | Crime | Yes | 20 | 15 | 1,4 |
| | Terrorism | Yes | 9 | 6 | 1,6 |
| If we could spend more money on fighting crime | Build new prisons | Yes | 15 | 11 | 1,5 |
| | Recruit more police officers | Yes | 20 | 17 | 1,3 |
| The authoritarian and attitudes toward law and order | Schools should teach children to obey authority | Fully agree | 21 | 15 | 1,5 |
| | Young people today don't have enough respect for traditional French values | Fully agree | 17 | 14 | 1,3 |
| | Young people today don't have enough respect for traditional French values | Rather agree | 60 | 44 | 1,9 |
| | People who break the law should be given stiffer sentences | Fully agree | 36 | 25 | 1,7 |
| The main cause of crime | Poverty | Yes | 13 | 22 | 0,5 |
| | The lack of family support | Yes | 50 | 41 | 1,5 |

The column 'class' corresponds to the proportion of respondents concerned from the group (%), the column 'sample' to the proportion of respondents concerned in the general population (%), and the 'odds ratio' column represents the value of the odds ratio relative to the difference between these two proportions.

are overrepresented among respondents who believe that the sense of belonging to the community and the sense of right and wrong amongst neighbors *have decreased*. But are also overrepresented in this group those who believe, on the contrary, that the sense of belonging to the community and the sense of shared values of the inhabitants *have increased*. In addition, the behavior of young people in the neighborhood – which is of great concern to the 'worried-dysfunctional' – does not characterize the relationship to the residential environment of the 'worried-functional' [On this question, the 'worried-functional' are not significantly different from the population as a whole.] (Table 12).

There is a categorical difference between the two groups in terms of moral and political values. When asked about the most important problem in French society, the 'worried-functional' mainly indicate *health* rather than *crime* as the priority problem. In general, these respondents have the same model of representations as the 'unworried'. They also overwhelmingly advocate *youth prevention* to combat crime and are less favorable than the average to *building new prisons*. 'Worried-functional' are also more likely to cite *poverty* as the main cause of crime. This group also shows the same tendency as the 'unworried' to distance themselves from authoritarian attitudes. There is an over-representation in this group of respondents who *rather disagree* that people who did not respect authority should be given stiffer sentences, and that young people today don't have enough respect for traditional French values. 'Worried-functional' are also less likely to *fully agree* with this first statement and more likely to *fully disagree* with the second (Table 13).

**Table 11. Perceived likelyhood, victimization and worry about crime (worried-functional).**

| Latent variable | Manifest variable | Modalities | Class (%) | Sample (%) | Odd Ratio |
|---|---|---|---|---|---|
| Perceived likelyhood of being victimized | Burglary | Quite likely | 20 | 12 | 1,8 |
| | Burglary | Quite likely | 18 | 9 | 2,1 |
| | Larceny | Quite likely | 28 | 11 | 3,0 |
| | Physical assault | Quite likely | 15 | 8 | 2,1 |
| | Physical assault | Unlikely | 56 | 40 | 1,9 |
| | Sexual assault | Quite likely | 9 | 3 | 3,5 |
| | Sexual assault | Unlikely | 43 | 27 | 2,0 |
| | Verbal assault | Quite likely | 50 | 19 | 4,2 |
| Experienced victimization | Burglary | Once | 11 | 4 | 2,6 |
| | Damage of vehicle | Once | 26 | 12 | 2,6 |
| | Damage of the housing | Once | 9 | 3 | 2,7 |
| | Insults | Once | 44 | 20 | 3,2 |
| | Insults | More than once | 19 | 7 | 3,1 |
| | Physical assault | Once | 7 | 4 | 1,7 |
| | Threats | Once | 29 | 12 | 2,9 |
| | Threats | More than once | 11 | 4 | 3,3 |
| Worry about crime | Burglary | Once | 29 | 13 | 2,7 |
| | Burglary | More than once | 32 | 12 | 3,5 |
| | Physical assault | Once | 20 | 8 | 2,9 |
| | Physical assault | More than once | 22 | 5 | 5,4 |
| | Sexual offense | Once | 8 | 3 | 2,9 |
| | Sexual offense | More than once | 15 | 3 | 5,5 |
| | Larceny | Once | 20 | 7 | 3,3 |
| | Larceny | More than once | 30 | 8 | 5,0 |
| | Vehicle theft | Once | 21 | 9 | 2,7 |
| | Vehicle theft | More than once | 28 | 9 | 4,1 |
| | Verbal assault | Once | 35 | 18 | 2,4 |
| | Verbal assault | More than once | 34 | 10 | 4,4 |

The column 'class' corresponds to the proportion of respondents concerned from the group (%), the column 'sample' to the proportion of respondents concerned in the general population (%), and the 'odds ratio' column represents the value of the odds ratio relative to the difference between these two proportions.

To sum up, 'worried-functional' are defined by a higher level of exposure to the risk of victimization and more frequent worries about crime than the national average. They are more likely to perceive problems and disorders in their neighborhoods. However, their daily experiences of crime and disorders does not negatively influence their entire system of representations as is the case for members of the 'worried-dysfunctional' group. These respondents have no concerns about crime and their moral and political values are distant from authoritarian attitudes.

### Results of the regression analysis

The previous analysis presents four classes, each based on a specific relationship structure. To ensure that these combinations of variables are grounded in concrete social realities, we performed a logistic regression analysis. The classes are placed as dependent variables and the socio-demographic predictors as independent variables. The objective is to identify the social groups associated with each of these four classes. The results of the analysis are presented in Table 14.

**Table 12. Perception of local environment (worried-functional).**

| Latent variable | Manifest variable | Modalities | Class (%) | Sample (%) | Odd Ratio |
|---|---|---|---|---|---|
| The attitudes toward social change in the community | A sense of belonging to the community | Had increased | 19 | 14 | 1,5 |
| | A sense of belonging to the community | Had decreased | 29 | 24 | 1,3 |
| | A sense of shared values amongst people who live here | Had increased | 14 | 10 | 1,5 |
| | A sense of right and wrong amongst people who live here | Had decreased | 39 | 31 | 1,4 |
| The concern about social cohesion | People in this area are trutworthy | Yes | 21 | 30 | 0,6 |
| | This area has a close, tight-knit community | Yes | 4 | 8 | 0,4 |
| The perception of problems in neighborhood | Bad reputation | Yes | 12 | 8 | 1,6 |
| | Crime | Yes | 29 | 18 | 1,8 |
| | Drug | Yes | 28 | 22 | 1,4 |
| | Family violence | Yes | 19 | 10 | 2,2 |
| | Noise | Yes | 27 | 20 | 1,4 |
| | Youth gangs | Yes | 19 | 13 | 1,6 |

The column 'class' corresponds to the proportion of respondents concerned from the group (%), the column 'sample' to the proportion of respondents concerned in the general population (%), and the 'odds ratio' column represents the value of the odds ratio relative to the difference between these two proportions.

**Table 13. Social and political attitudes (worried-functional).**

| Latent variable | Manifest variable | Modalities | Class (%) | Sample (%) | Odd Ratio |
|---|---|---|---|---|---|
| If we could spend more money on fighting crime | Build new prisons | Yes | 7 | 11 | 0,6 |
| | Develop youth prevention | Yes | 73 | 68 | 1,3 |
| Priority problem in French society | Crime | Yes | 10 | 15 | 0,6 |
| | Health | Yes | 28 | 22 | 1,4 |
| The authoritarian and attitudes toward law and order | People who break the law should be given stiffer sentences | Rather disagree | 19 | 15 | 1,3 |
| | People who break the law should be given stiffer sentences | Fully agree | 13 | 25 | 0,4 |
| | Young people today don't have enough respect for traditional French values | Fully disagree | 11 | 8 | 1,4 |
| | Young people today don't have enough respect for traditional French values | Rather disagree | 41 | 34 | 1,3 |
| The main cause of crime | Poverty | Yes | 33 | 22 | 1,7 |

The column 'class' corresponds to the proportion of respondents concerned from the group (%), the column 'sample' to the proportion of respondents concerned in the general population (%), and the 'odds ratio' column represents the value of the odds ratio relative to the difference between these two proportions.

The most contributing factors to the group of 'unworried' are age, type of housing and region of residence. It is first of all the oldest class on average. All else being equal, individuals aged 35–39, 60–64, and 70 years and older are more likely to belong to this group than those aged 18–24. Residing in a house outside urban area is also a predictive factor for belonging to this group compared to residing in a relegated neighborhood. Moreover, the 'unworried' are more likely to live in Western France than in the Paris region compared to the rest of the French population. Let us specify that this geographical area includes the regions (Bretagne, Normandy, Pays de Loire) where income inequality is the lowest in the country [87]. Finally, on the declared happiness scale, the 'unworried' rank higher than the national average.

**Table 14. Logistic regression model.**

| Predictors | Unworried | | Worried-dysfunctional | | Anxious | | Worried-functional | |
|---|---|---|---|---|---|---|---|---|
| | OR | p | OR | p | OR | p | OR | p |
| (Intercept) | **0.09** | ** | **0.49** | * | 0.60 | | 0.74 | |
| *sex* | | | | | | | | |
| [Men] | - | | - | | - | | - | |
| Women | 1.08 | | 1.10 | | **0.70** | ** | **1.50** | * |
| *age* | | | | | | | | |
| [18_24] | - | | - | | - | | - | |
| 25_29 | 1.67 | | 0.55 | | 1.02 | | 0.66 | |
| 30_34 | 1.94 | | 0.51 | | 0.72 | | 0.81 | |
| 35_39 | **2.70** | * | 1.23 | | 0.50 | | **0.29** | * |
| 40_44 | 2.44 | | 0.87 | | 0.52 | | 0.62 | |
| 45_49 | 1.98 | | 1.49 | | 0.53 | | 0.59 | |
| 50_54 | 2.20 | | 1.10 | | 0.69 | | 0.42 | |
| 55_59 | 1.71 | | 1.17 | | 0.92 | | 0.41 | |
| 60_64 | **3.01** | * | 0.74 | | 0.69 | | **0.26** | * |
| 65_69 | 2.21 | | 0.82 | | 0.70 | | 0.50 | |
| 70 years and over | **2.77** | * | 1.08 | | 0.66 | | **0.22** | ** |
| *Level of education* | | | | | | | | |
| [No degree] | - | | - | | - | | - | |
| Vocational-certificate | 0.92 | | 1.52 | | 0.95 | | 0.85 | |
| High school degree | 1.07 | | 1.12 | | 0.79 | | 1.23 | |
| 2 years of higher education | 1.23 | | 1.06 | | 0.75 | | 1.02 | |
| 5 years of higher education | 1.44 | | 1.09 | | **0.50** | ** | 1.27 | |
| *Standard of living* | | | | | | | | |
| [Can't do it debt free] | - | | - | | - | | - | |
| With difficulty | 1.44 | | 0.34 | | 1.15 | | 1.55 | |
| Barely | 1.95 | | 0.40 | | 1.15 | | 0.78 | |
| It is okay | 2.42 | | **0.25** | * | 1.04 | | 0.89 | |
| Comfortable | 2.45 | | **0.22** | ** | 1.19 | | 0.73 | |
| Very comfortable | 2.80 | | 0.26 | | 1.18 | | 0.37 | |
| *Dwelling type* | | | | | | | | |
| [Houses outside urban areas] | - | | - | | - | | - | |
| Houses in housing estate | 0.83 | | 1.45 | | 0.96 | | 1.31 | |
| Buildings in town | 0.77 | | **3.98** | *** | **0.56** | * | 1.15 | |
| Relegated neighborhood | **0.34** | ** | **3.88** | ** | 1.39 | | 1.09 | |
| Mixed dwellings | 0.84 | | **2.03** | ** | 0.67 | | 1.70 | |
| *Region* | | | | | | | | |
| [Greater Paris region] | - | | - | | - | | - | |
| The Paris Basin | 1.39 | | 0.90 | | 1.29 | | **0.36** | ** |
| North | 1.00 | | 0.85 | | 1.64 | | 0.53 | |
| East | 0.91 | | 0.81 | | **1.80** | * | 0.58 | |

*(Continued)*

Table 14. (Continued)

| | Unworried | | Worried-dysfunctional | | Anxious | | Worried-functional | |
|---|---|---|---|---|---|---|---|---|
| West | **1.62** | * | 0.71 | | 1.16 | | **0.39** | ** |
| Southwest | **1.56** | * | 0.68 | | 0.93 | | 0.66 | |
| Centre East | 1.16 | | 0.88 | | 1.41 | | **0.52** | * |
| Mediterranean | 1.25 | | 1.05 | | 1.28 | | **0.43** | * |
| Happiness | **1.08** | * | **0.90** | * | 1.02 | | 0.91 | |
| Observations | 1394 | | 1394 | | 1394 | | 1394 | |
| R2 Tjur | 0.053 | | 0.062 | | 0.044 | | 0.054 | |

This table presents the Odds Ratios and p-values of sociodemographic variables for the 'unworried', the 'worried-dysfunctional', the 'anxious', and the 'worried-functional' groups. Asterisks indicate levels of statistical significance: $p < .05$ (*), $p < .01$ (**), and $p < .001$ (***). No asterisk denotes that the association is not statistically significant.

The 'worried-dysfunctional' differ from the rest of the population by their standard of living and the type of housing occupied. Declaring the need to go into debt to make ends meet significantly increases the chances of belonging to this group compared to declaring no financial difficulties. Unsurprisingly, residing in densely populated urban environments, especially in disadvantaged neighborhoods composed of housing projects, is a predictive factor for belonging to this group compared to living in a non-urban house. Consistent with these results, several studies show that economic precariousness [30,88–90] and residing in a neighborhood with weak social, economic, and structural characteristics [91] are predictors of fear of crime. It may come as a surprise that the effects of gender and age are absent in this group. However, Jackson [4] finds a similar result: he shows that the effects of authoritarian attitudes and perceiving the neighborhood as declining (expressive fear) completely absorb the effects of gender and age on the perception of victimization risk and worries about crime (experiential fear). Finally, as expected [92], this group reports a lower level of happiness than the national average. While we cannot identify fear of crime as the cause of a decrease in quality of life as Gray et al. [2] do, this low level of happiness, combined with high experiential fear and high expressive fear, leads us to describe this group as 'worried-dysfunctional'.

Sex, level of education, type of housing and region of residence are the main contributing factors of the 'anxious'. In this group, being a man compared to being a woman, being a tenant or homeowner of houses outside urban areas compared to being a tenant or homeowner of urban buildings, having no diploma compared to having a high level of education (5 years or higher education), and living in the Eastern Region (which is particularly affected by deindustrialization and in which the unemployment rate is relatively high) compared to living in the Paris Region significantly increase the chances of belonging to this group rather than to the other three. These results confirm the findings of several French studies. Robert and Pottier [81], Robert and Zauberman [93], Zauberman et al. [29], and Jardin et al. [26] show that low educational attainment, right-wing political alignment, belonging to the working and lower-middle classes, and residing in rural or semi-rural areas are strong determinants of concern about crime. These sociodemographic characteristics thus appear as explanatory factors for expressive fear (social and political attitudes), but not for experiential fear.

Finally, the 'worried-functional' differ from the rest of the population by sex, age and region of residence. Being a woman compared to being a man, being between the ages of 18 and 24 compared to being between 35 and 39, 60 and 64, or over 70, and residing in the Paris Region compared to living in the Paris Basin, the West, the Central East, and the Mediterranean significantly increase the chances of belonging to this group. These results corroborate those of many studies. In the absence of a high level of expressive fear, being a woman is one of the strongest determinants of worry about crime [21,30]. Regarding the fact that fear of crime is more common among younger people compared to older

individuals, several studies since Ferraro's analyses [7] have supported this view [12,23,94,95]. Furthermore, Ceccato [96,97] shows that the fear of becoming a victim of crime is higher in densely populated urban areas than in medium-sized cities and rural areas. In light of the results of our analysis, it is worth noting that these factors contribute to explaining the experiential aspect of fear of crime, rather than its expressive aspect. Finally, regarding the declared level of happiness, these respondents do not differ from the national average. In light of this result, it seems appropriate to refer to them as 'worried-functional'.

## Discussion

Fig 2 below presents the model extracted from our analyses. This graph shows the distribution of the 4 classes along two axes: experiential fear (on the x-axis) and expressive fear (on the y-axis). The classes located on the left side of the graph (the 'anxious' and the 'unworried') are characterized by a low level of experiential fear, while those located on the right (the 'worried-dysfunctional' and the 'worried-functional') are characterized by a high level of experiential fear. Additionally, the classes located at the bottom of the graph (the 'unworried' and the 'worried-functional') are characterized by a low level of expressive fear, while those located at the top (the 'anxious' and the 'worried-dysfunctional') are characterized by a high level of expressive fear.

These results confirm our first hypothesis (H1), according to which there are relationship structures that strongly and consistently associate *experiential fear* and *expressive fear*. Thus, for the 'unworried', the lack of concern about crime or social change in the community, low propensity for authoritarian attitudes and low perception of disorder are associated with a very low perceived likelihood of victimization and extremely infrequent worries. Conversely, the 'worried-dysfunctional' are highly exposed to the risk of victimization, more frequently afraid than average, very concerned about crime, very sensitive to authoritarian attitudes and highly dissatisfied with their neighborhood and relationships with neighbors. These results corroborate those of Jackson [4] and Farrall et al. [1], who, for the first time, link the components of experiential and expressive fears in a linear model. Similarly, in the second version of the EEF model, the classes with the same names – the 'unworried' and the 'worried-dysfunctional' – exhibit exactly the same characteristics as ours [2].

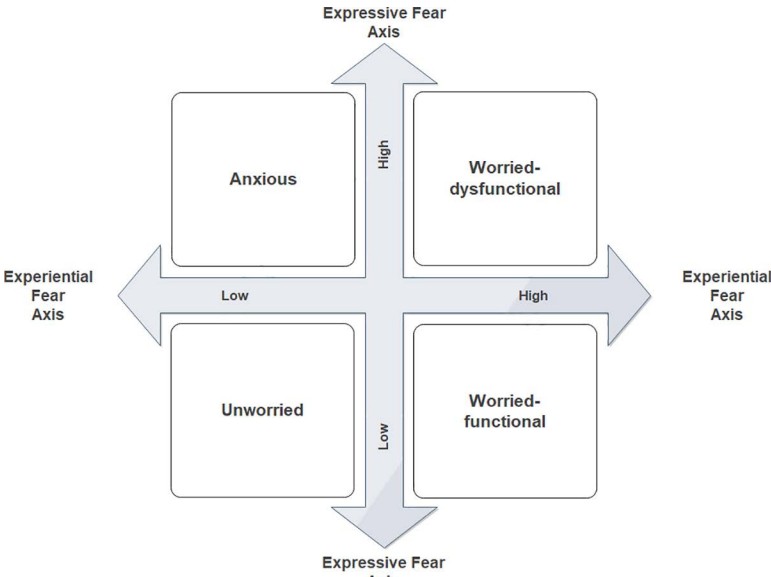

**Fig 2. Distribution of the 4 classes around the two axes of experiential and expressive fear of crime.**

But our results do not only confirm this initial relationship structure; they go beyond it. Indeed, in addition to these first two classes, our analysis adds two more. As our second hypothesis (H2) suggests, these classes are characterized by a dissociation between *experiential fear* and *expressive fear*. Consequently, the variations with the previous classes are no longer of degree (different classes sharing the same structure of relations with variations following a linear logic) but of kind (classes with different structures of relations).

Thus, the 'anxious' display strong concern about crime and pronounced authoritarian attitudes regarding low and order. Despite this, they say they are relatively satisfied with their living environment, not much confronted to victimization and only very rarely worried about crime. The configuration specific to this class reveals several new findings:

- the 'anxious' assess the likelihood of being a victim of an offense as lower than the average (this result contrasts sharply with the first version of the EEF model, which found a significant positive link between these two variables);

- the 'anxious' do not differ from the rest of the population regarding the perception of disorders in their neighborhood (contrary to the results presented in the first two versions of the EEF model, which established a significant positive link between these two variables);

- the 'anxious' display concerns toward social change in the community, coupled with pronounced authoritarian and security-oriented attitudes (contrary to the results presented in the first two versions of the EEF model, which establish no direct link between the components of expressive fear and anxiety).

Thus, the 'anxious' not only differ from the 'worried' (both functional and dysfunctional) by having a lower risk of victimization ["BY CONTRAST, WE FOUND THAT ANXIETY ABOUT CRIME WAS LESS RELATED TO THE REALITIES OF A DAILY LIFE AFFECTED BY CRIME. NEVERTHELESS, THESE INDIVIDUALS CONTINUED TO BE CONCERNED ABOUT DISORDER AND SOCIAL STABILITY. PARTICIPANT WERE, AKIN TO THE WORRIED, LIKELY TO LINK PHYSICAL CUES TO PROBLEMS OF DECLINING QUALITY BONDS AND SOCIAL COHESION, PERCEIVED DISORDER, AND RISK […]. HOWEVER, THE ANXIOUS WERE DISTINGUISHABLE FROM THE WORRIED, IN THAT THE 'EXPERIENTIAL' INDICATORS OF CRIME AND VICTIMIZATION WERE SIGNIFICANTLY LOWER THAN WITH WORRIED" [1:PP.228].], but also by a lack of perceived likelihood and lower concerns regarding collective efficacy, social cohesion, and disorders in their neighborhood [TO BETTER SET APART THE 'ANXIOUS' FROM THE 'WORRIED', THE SAME RESEARCHERS ADD A FEW LINES LATER THAT "THE 'ANXIOUS' FELT LESS AT RISK; WERE LESS CONCERNED ABOUT DISORDER" THAN THE 'WORRIED' [1:PP.228]. HOWEVER, THIS INTERPRETATION DOES NOT SUPPORT THEIR RESULTS. REGARDING THE PERCEPTION OF DISORDER, THE FIRST VERSION OF THE MODEL APPLIED TO THE LOCAL SURVEY SHOWS AN EQUIVALENT EFFECT OF THIS VARIABLE ON ANXIETY (.10*) AND WORRY (.12*) [1:PP.226]. APPLIED TO THE DATA FROM THE BRITISH CRIME SURVEY (BSC), THE PERCEPTION OF DISORDER EVEN HAS A STRONGER EFFECT ON ANXIETY (.25*) THAN ON WORRY (.14*) [1:PP.221]. THE RESULTS FROM THE SECOND VERSION OF THE MODEL DO NOT FURTHER CONFIRM THE INTERPRETATION PROPOSED BY THE RESEARCHERS. HERE, THE ODDS RATIOS VARY BETWEEN 1.19* AND 1.22* FOR THE EFFECT OF THIS VARIABLE ON EACH OF THE FOUR CLASSES, INDICATING VERY SMALL DIFFERENCES [2:PP.83]. THE PROBLEM IS EXACTLY THE SAME FOR THE PERCEIVED LIKELIHOOD OF BEING A VICTIM. THE FIRST VERSION OF THE MODEL APPLIED TO THE BCS SHOWS A STRONGER EFFECT OF THIS VARIABLE ON WORRY (.53*) THAN ON ANXIETY (.41*), BUT THE RESULTS FROM THE LOCAL SURVEY INDICATE THE OPPOSITE. THE STANDARDIZED COEFFICIENT IS HIGHER FOR THE RELATIONSHIP BETWEEN PERCEIVED LIKELIHOOD AND ANXIETY (.62*) THAN FOR THE RELATIONSHIP BETWEEN THIS SAME VARIABLE AND WORRY (.40*). THUS, BASED ON THESE RESULTS, IT APPEARS THAT ONLY VICTIMIZATION AND WITNESSING AN OFFENSE DISTINGUISH THE 'WORRIED' FROM THE 'ANXIOUS'.]. This result seems quite consistent. In the different versions of the EEF model [1,2,4], it is unclear why the same causes – perception of victimization risk and perception of disorder in the neighborhood – produce different effects (anxiety or worry). In contrast, our results suggest that the causes diverge significantly and extend well beyond the experience of victimization alone. If the 'anxious' do not shift into the worried groups, it may be precisely because these respondents perceive a low level of physical and social disorder and criminal activities in their residential environment [26,29,80,81]. In contrast, the 'anxious' are less satisfied with their neighborhood than the 'unworried' due to a higher level of concern about the long-term social change in the community. But this anxiety is less based on the perception of signal crime and disorder in the neighborhood (which would directly shift the 'anxious'

to the 'dysfunctional-worried') than on concerns about the evolution of society and one's place within it [98]. Pointing to crime as the origin of societal problems is a way to transform this diffuse anxiety, which is difficult to identify, name, and even manage, into a specific problem that is identifiable, nameable, and manageable [3]. Thus, for this class, our results corroborate those of Hirtenlehner and Farrall [3], which show that anxiety about crime is correlated with economic and social concerns, and strongly contrast with those of Gray et al. [2], who found no direct link between anxiety about crime, concern about moral decline, and concern about long-term social change in the community.

Finally, our results reveal the existence of a fourth, unprecedented class that is completely opposed to the 'anxious' group. The 'worried-functional' are more exposed than the average to attacks and are more frequently worried. They are also more likely to highlight problems in their neighborhoods. However, these respondents are not concerned about crime and do not have authoritarian attitudes. The configuration specific to this class opposes the main result of the second and third versions of the EEF model:

- According to Hirtenlehner and Farrall [3], social and moral concerns are the main predictors of worry/anxiety (indistinctly). On the contrary, our results show that there is a group (the 'worried-functional') whose worry about crime is not associated with social and moral concerns.

- According to Gray et al. [2], concern about moral decline is a predictor of belonging to the 'worried-functional' group. On the contrary, our results show that, in this specific case, worry about crime is not associated with a declinist attitude toward social relations [23].

Thus, our model not only distinguishes the 'worried' from the 'anxious', but it also clearly distinguishes the 'worried-functional' from the 'worried-dysfunctional'. According to Gray et al. [2], these two groups share two contributing factors: the perception of disorder and concern about moral decline. In contrast, collective efficacy and victimization are predictors of the 'worried-dysfunctional' group only. Our analyses reveal significantly different results. In both cases, worry about crime is associated with high exposure to the risk of victimization and a high perceived likelihood of being a victim. Both the 'worried-functional' and the 'worried-dysfunctional' are also more likely than average to hold negative judgments about their residential environment. Both groups perceive more disorder and are more concerned about social cohesion in their neighborhood (although this latter variable is more strongly associated with the 'worried-dysfunctional' than with the 'worried-functional'). Several studies have shown that victimization [7], risk perception [7,99], and perceived incivilities [100,101] are predictors of concrete fears (worry about crime) rather than abstract fears (anxiety/concern about crime) [102]. It is therefore consistent that both worried groups share these same characteristics. In contrast, like Gray et al. [2], we find that concern about collective efficacy is associated only with the 'worried-dysfunctional' group. The structure of relationships characterizing the 'worried-dysfunctional' corroborates the results of Brunton-Smith et al. [103]. According to these researchers, individuals who perceive a high level of collective efficacy correspondingly have a lower level of worry about crime than those who perceive a low level of collective efficacy. However, the linear relationship between these variables disappears for the 'worried-functional' group. In this case, worry about crime, perception of disorder, and victimization are no longer closely associated with collective efficacy. This result supports evidence from Kuen et al. [104], who find no link between fear of crime and collective efficacy once the perception of disorder is controlled for. The distinction between the two groups of the 'worried' thus seems particularly relevant for understanding the nuanced findings in the literature regarding the link between worry about crime and collective efficacy. But what clearly distinguishes the 'worried-functional' from the 'worried-dysfunctional' is their social and political attitudes and values. While the 'worried-dysfunctional' are highly concerned about crime and display pronounced authoritarian attitudes, the 'worried-functional' do not share these ideological positions. This result sharply contrasts with those of the second version of the EEF model, which find a significant link between concern about moral decline and worry about crime (both functional and dysfunctional without out distinction).

Ultimately, the main contribution of our analysis is the identification of four contrasting configurations of the relationship between experiential fear and expressive fear, whereas the first [1,4] and third [3] versions of the EEF model identify only part of these structures (the 'unworried' and the 'worried-dysfunctional' in the first version, and the 'anxious' in the third). Meanwhile, the second version of the EEF model [2] presents much less distinct combinations of variables. Table 15 summarizes the main contributions of our model compared to the three previous ones.

At this stage of analysis, we could argue that these classes are nothing more than abstract concepts that are not rooted in any tangible social reality. But the results of our regression analysis show the opposite. These classes take shape within specific populations, characterized by various sociodemographic factors that reflect different forms of inequalities in relation to fear of crime. This result validates our third hypothesis (H3), which states that sociodemographic predictors vary significantly from one class to another.

Being over 60 is a strong predictor of belonging to the 'unworried' group. Now, economic wealth significantly increases over a lifetime: individuals over 60 possess, on average, 13 times more than those under 30 [105]. More than others, the 'unworried' group have the economic and financial means to protect themselves from crime. Therefore, it is not surprising that residing in regions with the most income equality [On the link between crime and income inequality, see particularly Coccia [106] and Coccia and Cohn [107].] and living in individual houses outside urban areas are predictors of belonging to this group. These favorable socioeconomic conditions help distance individuals from potential risks of victimization (low levels of experiential fear) as well as concerns about crime (low levels of expressive fear).

The 'worried-dysfunctional' group have a radically different social profile. Economic precariousness and residing in disadvantaged neighborhoods are both significantly predictive of belonging to this class [30,88–90]. These living conditions help to better understand the high levels of experiential fear and expressive fear displayed by the 'worried-dysfunctional', who are more exposed to the risks of theft, burglary and assault and are less able to cope with them, at least financially. In the absence of other options, demands for increased authority and security likely appear as the sole recourse to improve living conditions deemed critical and from which one cannot escape [108].

Low educational attainment, residing in regions heavily affected by deindustrialization, and not living in urban areas significantly influence belonging to the 'anxious' group. These factors explain, at least in part, this group's specific relationship with experiential fear (low level) and expressive fear (high level). Indeed, the 'anxious' perceive a low level of signal crime and signal disorder in their residential neighborhoods [38,39]. In contrast, they seem to have the greatest difficulty in adapting to societal changes [55,109]. This group's obsession with law, order and crime-related safety seems to reflect the anxiety about social decline much more than worry about personal or property harm. In other words, authoritarian attitude and concern about crime

**Table 15. Comparative summary of the results of the new model with the three previous versions of the EEF model.**

|  | Results of the previous models | Results of the present model (comparative) |
|---|---|---|
| Model EEF Version 1 | . Social and political attitudes → perceptions of the neighborhood → worry and anxiety | . Structure found with the 'unworried' and 'worried-dysfunctional' groups<br>. Discovery of two complementary and distinct structures (the 'anxious' and 'worried-functional' groups) |
| Model EEF Version 2 | . Neighborhood disorder perception → worry and anxiety<br>. Social and political attitudes → worry (but not anxiety) | . Neighborhood disorder perception characterizes the 'worried' groups (but not the 'anxious' group)<br>. Social and political attitudes characterize the 'worried-dysfunctional' and 'anxious' groups (but not the 'worried-functional' group) |
| Model EEF Version 3 | . Social and political attitudes → worry/anxiety<br>. Negligible effect of neighborhood disorder perception on worry/anxiety | . Social and political attitudes characterize the 'worried-dysfunctional' and 'anxious' groups (but not the 'worried-functional' group)<br>. Negligible effect of neighborhood disorder perception on the 'anxious' group (but not on the two 'worried' groups) |

The first column of the table presents the main results from the three previous versions of the EEF model, while the second column displays the main results of the new version. The symbol "→" indicates an influence relationship and shows the direction of the relationship.

rather appear here as a form of social reaction to perceiving one's own position in the social hierarchy as fragile and threatened [26,29,81]. In this sense, it is interesting to note that the 'anxious' exhibit similar social characteristics to those of voters for the Rassemblement National (an extreme right party in France that has been achieving increasingly significant results for about fifteen years) [FEW STUDIES FOCUS ON THE RELATIONSHIP BETWEEN FEAR OF CRIME AND VOTING FOR FAR-RIGHT PARTIES. THIS IS DUE TO THE FACT THAT VERY FEW SURVEYS INCLUDE QUESTIONS ON BOTH FEAR OF CRIME AND ELECTORAL CHOICES IN THE SAME QUESTIONNAIRE. AS A RESULT, MOST STUDIES FOCUS MORE ON THE RELATIONSHIP BETWEEN POLITICAL ATTITUDES AND FEAR OF CRIME, BUT WITHOUT ELECTORAL DATA ON ONE SIDE [110,111], AND ON VOTING AND CRIME (POLICE DATA), BUT WITHOUT DATA ON FEAR OF CRIME ON THE OTHER SIDE [112,113].] [110,111]. Less educated than average, these voters are – like the 'anxious' – overrepresented among residents of rural/ semi-rural areas and regions affected by deindustrialization [112,114]. They are also more concerned about crime, even though they are less exposed to victimization and are less frequently worried about crime than the average [115].

Finally, the results of the regression analysis seem to indicate – indirectly – that the high level of experiential fear of the 'worried-functional' is due more to their lifestyles than to unfavorable living conditions, as is the case with the 'worried-dysfunctional'. Indeed, individuals aged 18–25 and women (two predictors of belonging to the 'worried-functional' group) are the primary users of public transportation [23,116]. Furthermore, individuals aged 18–25 and Paris region residents (the third predictor of belonging to the 'worried-functional' group) are more likely to travel at night using public transportation than older individuals and residents of other French regions [117]. These same social groups also engage in nighttime activities more regularly [118]. Now, these contexts (the use of public transportation, voluntary nighttime activities) are often cited as the most threatening, particularly by young women who are more worried than others about being victims to sexual assaults in these situations [23,27,28,45,116,119,120]. Nevertheless, we can suppose that it is likely easier – albeit still challenging – to manage the exposure to the risk of victimization related to routine activities (particularly voluntary nighttime activities) than the exposure to risk associated with one's living environment [9,121,122]. This difference with the 'worried-dysfunctional' likely plays an important role in explaining the low level of expressive fear of the 'worried-functional'. The absence of authoritarian and security-oriented attitudes could be explained by a more circumstantial exposure to the risk of victimization, "delimited" to certain spaces and specific locations [123]. By losing its random and systematic nature, the risk of victimization could thus generate a high level of experiential fear, without necessarily leading to a high level of expressive fear [THIS HYPOTHESIS ALSO PROVIDES AN EXPLANATION FOR THE WEAK LINK BETWEEN WORRY ABOUT CRIME AND COLLECTIVE EFFICACY THAT CHARACTERIZES THE 'WORRIED-FUNCTIONAL' GROUP. IF EXPOSURE TO RISK IS RELATED (AT LEAST IN PART) TO ROUTINE ACTIVITIES THAT TAKE PLACE OUTSIDE THEIR NEIGHBORHOODS, IT EXPLAINS WHY COLLECTIVE EFFICACY PLAYS A LIMITED ROLE IN REDUCING WORRY ABOUT CRIME.].

## Limitations

The present study, nonetheless, has certain limitations. The first one is related to the distinction between *anxiety*, as measured by Farrall et al. [1], and *concern about crime*, as measured in French victimization surveys. While both concepts refer to "abstract fear" differentiated from "concrete fear" (experiential fear), they are measured differently. Concern about crime involves placing crime as the primary problem in society [79], whereas anxiety involves declaring being afraid of crime without having experienced any worry about crime in the past 12 months [1]. It is therefore possible that these two concepts measure slightly different emotions. However, comparing results between the two surveys shows the similarity of these two concepts. Like concern about crime, anxiety is not directly related to the experience of victimization or even the worry about crime. If differences exist, they should not be significant enough to challenge the comparison of the two models.

The second limitation of this study is related to the small $R^2$ values for the regression models, ranging between 0.041 and 0.065. The most likely explanation lies in the central role of the variables integrated into the hierarchical cluster analysis. Jackson [4], for example, demonstrates that social and political attitudes and neighborhood perceptions questions entirely absorb the effect of several sociodemographic variables on worries about crime. More generally, sociodemographic variables have always been very insufficient in explaining fear of crime. This is why, over the past 50 years,

researchers have worked tirelessly to develop and test other models (the victimization model, the disorder perception model, the perceived vulnerability model, etc.). To increase the explanatory power of the model, it should not be limited to sociodemographic variables. It would be appropriate to test other factors that the scientific literature shows contribute to explaining fear of crime, such as health status [2], perceived physical vulnerability [34,124–127], perceived consequences of victimization [126,128], or routine activities [9,121,122,129,130]. That being said, our regression analysis was never intended to propose the best possible explanatory model, but rather to determine the main sociodemographic predictors of the 'unworried', 'worried-dysfunctional', 'anxious', and 'worried-functional' groups. In doing so, and despite the small $R^2$ values, our results reveal the specific effects of several independent variables, whereas the earlier versions of the EEF model showed no effects of sociodemographic factors (except for gender in the second version). In summary, this new version of the EEF model reintroduces the effects of several sociodemographic variables, even though these effects explain only a small portion of the variance between the classes. The question now is whether sociodemographic predictors could have a more significant effect than observed in the present study. We believe this could be the case, provided that much more precise sociodemographic data is used. As a reminder, our regression analysis was conducted on a sample of 1,394 individuals across mainland France. This sample is too small to conduct a detailed territorial analysis. Zauberman et al. [29] and Jardin et al. [26] demonstrate the relevance of working at the municipal level to precisely understand how fear of crime, victimization and concern about crime combine in different ways according to territories and their populations. More generally, several studies have highlighted the importance of studying fear of crime at the neighborhood level [41,43], the block level [131,132], and even at the street segment level [104]. It would be interesting to replicate the analysis presented in this article on larger-scale data to conduct a much finer geosocial analysis of the relationship with fear of crime. In these conditions, it is highly likely that sociodemographic variables contribute more to explaining membership in each of the four classes.

## Conclusion

Beyond these two limitations, this study has implications for research on fear of crime as well as for the implementation of public policies.

### Theoretical implications

Theoretically, our model offers two contributions. The first is to provide new insights into the relationship between the variables of the EEF model. Our hierarchical clustering analysis shows that some variables are more systematically associated than others. This is above all the case for worries about crime, victimization, perceived likelihood and the perception of disorders/crime in the neighborhood which strongly confirms Ferraro's model [7]. Thus, according to our results, these variables are the main components of *experiential fear*. This is subsequently the case of crime concerns, representations of crime, authoritarian and repressive attitudes (solutions put forward to combat crime) which in this sense are the pillars of *expressive fear* [133]. Perceptions of long-term social change in the community, collective efficacy and social cohesion in the neighborhood appear to act as intermediate variables between these two dimensions. They are evaluated positively when there are low levels of *experiential fear* and *expressive fear* (for the 'unworried') and negatively in the opposite configuration (the 'worried-dysfunctional') [1,4]. However this relationship becomes more complex in the case of the last two classes. The weak *expressive fear* of the 'worried-functional' seems to immunize them from a declinist vision of social relations in their neighborhoods. Nevertheless, these respondents do not express a very high level of satisfaction with community social life – mainly because of a relatively high level of *experiential fear*. In the case of the 'anxious', the low level of *experiential fear* is associated with a relatively high level of satisfaction with neighborhood relations, with the exception, however, of the young people's respect for others and for authority in the neighborhood considered, more than in the general population, in decline. This major difference leads to the transition to the strong *expressive fear* that characterizes this particular group. Thus, in the case of the 'anxious' and the 'worried-functional', the linear relationship – present

for the 'unworried' and the 'worried-dysfunctional' – between experiential fear, expressive fear, and concerns about the community disappears. At an equivalent level of victimization and worry about crime, the 'worried-functional' are less concerned about collective efficacy, social cohesion, and long-term social change in the community than the 'worried-dysfunctional'. Similarly, at comparable levels of victimization and worry about crime, the 'anxious' are much more concerned about social change in the community than the 'unworried'. Clearly then, the role played by representations of community social life in the relationship between *experiential fear* and *expressive fear* would benefit from further in-depth study in future research. Yet this objective could face major obstacles. Recent publications question the scientific relevance of many of these concepts. For Farrall, Jackson, Gray and Hirtenlehner, the fact that expressive fear influences experiential fear does not imply the interchangeability of the concepts that these two categories cover. However, this radical position was adopted by Etopio and Berthelot [134]. On the grounds that their respondents do not make a distinction in the use of terms of fear, anxiety, concern to evoke fear of crime, these authors argue that these concepts are interchangeable. They conclude that there is no distinction between *fear* and *concern* as Furstenberg [79] and Ferraro and LaGrange [68] could claim. Our results strongly refute this argument. For the 'worried-functional', worry about crime – a component of experiential fear – expresses itself independently of concern about crime – a component of expressive fear – and vice versa for the 'anxious'. That is why, on the contrary, it seems very important to encourage future research to clearly distinguish these different components of fear of crime. Even today, many studies measure this social phenomenon using insufficiently precise indicators [65,102]. As a result, it is impossible for these studies to differentiate between the various classes of the 'anxious' and the 'worried', who experience fear of crime in very different ways.

The second contribution of this analysis is to offer new insights into the role of sociodemographic factors. While these variables have been the subject of numerous studies, the results are often contrasting. Some studies show that older individuals are more fearful than younger ones [68,70,90], while others demonstrate the opposite effect [7,12,94]. The same applies to socioeconomic status [These debates are equally important regarding the role of race/ethnicity. However, as ethnic statistics are prohibited by law in France, we were unable to include this variable in our regression model.]. Some researchers establish a significant correlation between disadvantaged individuals and a high level of fear of crime [89,90,135], while others find no link between these two variables [120,128,136]. To explain these contrasting results, two major explanations have been proposed. The first relates to measuring instruments and the importance of capturing 'concrete fears' actually experienced in situations, rather than the abstract probability of being a victim of crime [7,73]. The second emphasizes the complexity of fear of crime and the need to add other factors. For instance, physical vulnerability serves as an intermediary variable between social vulnerability and fear of crime [34,128]. Our study adds an additional explanatory element. To understand the role of sociodemographic variables, it is crucial beforehand to identify different relationships to fear of crime, and more precisely to experiential fear and expressive fear. Thus, socioeconomic status predicts belonging to the 'worried-dysfunctional' groups but not to the 'worried-functional' and 'anxious' groups. Similarly, age contributes to explaining membership in the 'unworried' and 'worried-functional' groups but not in the 'anxious' and 'worried-dysfunctional' groups. Even gender, which is often considered the most consistent predictor of fear of crime, does not contribute to explaining membership in all groups. It predicts belonging to the 'anxious' and 'worried-functional' groups but not to the 'unworried' and 'worried-dysfunctional' groups. Therefore, there are different relationships to experiential fear and expressive fear structured around various groups with significantly varying sociodemographic characteristics. Hence the importance of identifying the configurations of fear of crime before attempting to determine the sociodemographic correlates.

## Policy implications

Another contribution of this analysis concerns the field of public policy. The different relationships to expressive fear and experiential fear call for the implementation of specific policies, actions and devices. Solutions drawn only from the repertoire of *situational crime prevention*, *crime prevention through environmental design* (CPTED) or *broken windows*

could have a positive effect for the 'worried-functional', moderate for the 'worried-dysfunctional', but probably zero for the 'anxious'. These hypotheses emerge from the results of our analysis. By focusing on the situation in which crime, disorder, or worries about crime arises, these crime prevention and control policies are more likely to impact experiential fear than expressive fear. That is why we hypothesize that these *crime prevention and control policies*, if implemented alone, will – at best – only be effective for the 'worried-functional'. In order to combat fear of crime of the 'worried-dysfunctional', it will likely be necessary to supplement this initial set of public policies with actions drawn from *community crime prevention* as well as the implementation of *social policies*. As we have seen, the 'worried-dysfunctional' face multiple difficulties. Even more so than the 'worried-functional' (for whom at least some experiential fears may be linked to routine activities), the 'worried-dysfunctional' hold very negative perceptions of their residential environment. *Community crime prevention* initiatives could help address neighborhood-related issues. At the same time, the implementation of *social policies* is essential to improve the financial conditions of this group, which is significantly more precarious than the average. As for the 'anxious', it is likely that none of these actions would have any effect on them. Members of this group are not particularly dissatisfied with their residential environment (which would justify *community crime prevention policies*); their living conditions are not worse than elsewhere (which would justify the deployment of *social policies*); and their level of experiential fear is very low (which would justify *crime prevention and control policies*). In the case of this group, it is actually more a matter of working on trust in social and political institutions, and more broadly of acting on the abstract – but no less intense – anxiety about losing social rank and about social decline. Paradoxically, the focus here should be less on tackling crime itself than on: (1) competition between social groups for access to state resources, and (2) relationships with social groups – particularly young people – who are perceived as threatening shared values and ways of life. Providing the most appropriate response, therefore, involves identifying precisely theses patterns and the associated sociodemographic groups. Ultimately, the main contribution of this article may be to demonstrate that fear of crime is not a dichotomous phenomenon ('worried'/ 'unworried'). This widely accepted conceptualization sets public policies up for failure by grouping individuals with very different relationships to fear of crime into an artificial category ('the worried').

### Ideas for future research

Developing this new version of the EEF model requires three preliminary conditions. The first is to systematically include all the variables of the EEF model (which, it should be recalled, are the main factors and components of fear of crime identified by uninterrupted research over the past fifty years) in victimization surveys. The second condition is to test this new version of the model in countries other than France. We hypothesize that the four classes identified in this article exist elsewhere, particularly in neighboring countries. The 'unworried' and the 'worried-dysfunctional' exhibit the same configurations in the United Kingdom [1,2,4], just as the 'anxious' do in Germany [3]. There is therefore no reason to believe that the French case is an exception. However, making conjectures for other countries, especially non-Western ones, is more challenging. In any case, to identify the full set of relationship structures (classes), we believe it is essential to use a mixed-methods approach rather than a SEM or a simple regression analysis. The third condition is to replicate this analysis on larger samples to allow for detailed geosocial analyses [26,137] and thus be able to precisely map the different patterns of fear of crime. With these three conditions met, the EEF model could become a highly valuable tool to help public authorities effectively tackle this multifaceted social phenomenon, which affects nearly half of the population, at least in France.

### Supporting information

**S1 Table. The variables integrated into the analysis as active variables.** The first column presents the three main categories of variables, the second lists the questions and their response modalities, and the third specifies the assctive variables included in the multivariate configuration analyses (Multiple Correspondence Analysis and Cluster Analysis). (DOCX)

**S1 File. Database SPIP2022.**

(ZIP)

## Author contributions

**Conceptualization:** Julien Noble, Antoine Jardin.

**Formal analysis:** Julien Noble, Antoine Jardin.

**Funding acquisition:** Julien Noble, Antoine Jardin.

**Methodology:** Julien Noble, Antoine Jardin.

**Writing – original draft:** Julien Noble, Antoine Jardin.

**Writing – review & editing:** Julien Noble, Antoine Jardin.

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
