## [Decision Letter · Decision Letter 0]

15 Jan 2025

Dear Dr. Noble,

Thank you for submitting your manuscript to PLOS ONE. After careful consideration, we feel that it has merit but does not fully meet PLOS ONE’s publication criteria as it currently stands. Therefore, we invite you to submit a revised version of the manuscript that addresses the points raised during the review process.

We look forward to receiving your revised manuscript.

Kind regards,

Ashraf Atta Mohamed Safein Salem

Academic Editor

PLOS ONE

Journal Requirements:

3. We note you have included a table to which you do not refer in the text of your manuscript. Please ensure that you refer to Tables 1-6 in your text; if accepted, production will need this reference to link the reader to the Table.

Additional Editor Comments:

1. Structural and Organizational Concerns:

- The authors are encouraged to amend the fragmented and confusing structure of the paper, they are recommended to completely reorganize the manuscript into clearly defined sections (e.g., Introduction, Theoretical Framework, Study Design, Results, Discussion, Conclusion).

- The excessive use of subheadings creates unnecessary complexity and redundancy, which detracts from the paper’s readability and coherence. Simplifying and synthesizing sections (e.g., merging theoretical framework and earlier versions of the model) is critical.

2. Methodological Clarity and Rigor:

- The methodology is unclear and requires restructuring into distinct subsections (Sample, Measures, Models, and Data Analysis). Additionally, specific elements like the logistic model equation and a consistent taxonomic criterion for group categorizations need to be clearly explained and justified.

- Authors should provide sufficient detail on effect sizes and the significance of sociodemographic predictors, which undermines the credibility of the results.

3. Presentation of Results:

- The presentation of results is dense and difficult to digest. Tables and figures are either unreadable or redundant (e.g., Table 6 with low R2R^2 and Figure 1). Authors are advised to consolidate results into fewer, more meaningful tables and moving supplementary details to an appendix is necessary.

- Authors are also recommended to synthesize key findings in a simple table within the discussion for better clarity.

4. Discussion and Implications:

- There is a need for a more critical synthesis and deeper analysis in the discussion. A concise explanation of results, highlighting unexpected or profound findings, and their relevance to existing literature.

- Practical and policy implications are underdeveloped, insufficient elaboration on public policy implications and generalizability of findings to other contexts.

5. Theoretical Framework:

- Authors are also encouraged to improve the theoretical framework by incorporating relevant suggested readings to provide a stronger foundation for the analysis. Additionally, previous versions of the model should be clearly synthesized into a cohesive narrative.

- The theoretical contribution of the paper is unclear without these improvements.

6. Limitations and Conclusions:

- Authors are advised to place limitations in the conclusion and focus on theoretical, policy, and future research implications. Subsections within the conclusion can improve clarity and emphasize the paper’s contribution to the field.

7. Scope and Generalizability:

- One of the main concerns is the lack of discussion on the generalizability of findings beyond the French context. Explicitly addressing how the findings may or may not apply to other cultural or geographical contexts will enhance the paper's broader relevance.

***Comments from PLOS Editorial Office:** We note that one or more reviewers has recommended that you cite specific previously published works. As always, we recommend that you please review and evaluate the requested works to determine whether they are relevant and should be cited. It is not a requirement to cite these works. We appreciate your attention to this request.*

Reviewers' comments:

Reviewer's Responses to Questions

**Comments to the Author**

1. Is the manuscript technically sound, and do the data support the conclusions?

Reviewer #1: Partly

Reviewer #2: Yes

2. Has the statistical analysis been performed appropriately and rigorously?

Reviewer #1: Yes

Reviewer #2: Yes

3. Have the authors made all data underlying the findings in their manuscript fully available?

Reviewer #1: No

Reviewer #2: Yes

4. Is the manuscript presented in an intelligible fashion and written in standard English?

Reviewer #1: Yes

Reviewer #2: Yes

Reviewer #1: Patterns of fear of crime. Revisit and synthesize the model of experiential and expressive fear of crime

The topics of this paper are interesting, though well known. The structure and content must be revised, and results have to be better explained by authors before to be reconsidered.

Title has to be shorter, clarify if this is a case study in France and the period.

Abstract has to be shorter, clarify the goal, what the study adds and implications for criminal justice system. Also here clarify if data are related to France and period under study.

The paper is long and has to be reduced. The structure is too fragmentated and confusing.

Authors have to structure the paper as follows.

-Introduction

-Theoretical framework

-Proposed model and Study design

-Results

-Discussion

-Conclusion

Authors have to avoid subheadings that create fragmentation and confusion. If necessary, can use bullet points (same comments for all sections).

Introduction has to better clarify the research questions of this study and provide more theoretical background about these topics, the socioeconomic drivers of crime in society, etc. After that they can focus on the topics of this study to provide a correct analysis for fruitful discussion. (See suggested readings that must be all read and used in the text).

Sections From the model's origins to its design, The results of the first version of the model, The results of the second version of the model, The results of the third version of the model can be merged and called “theoretical framework” reducing the content. A figure can synthetize these results of the previous version of the model to be clear for readers.

Section Hypothetico-Deductive Reasoning to Inductive Reasoning can be included in the study design. Reduce the section and provide a clear working hypothesis alternative or complementary to previous approaches.

The methods of this study is not clear. The section of Materials and methods must be re-structured with the following three sections and same order:

• Sample and data

• Measures of variables

• Models and Data analysis procedure. Logistic model has to be specified with equation.

Authors, again, have to avoid subheadings that create fragmentation and confusion. If necessary, you can use bullet points (same comments for section of results and all sections). A flow chart can show the logic of reasoning, indicating expected results (+ or -) to support then with empirical evidence and methods of inquiry to be clear for readers.

Categorization of groups (unworried, ‘worried-dysfunctional, anxious, etc.… hast to be clarified with a consistent taxonomic criterion underpinned in psychological literature.

Be synthetic, details in Appendix.

Results.

Figure 1 as done cannot be read and it is useless…it can be removed…

To reiterate, avoiding a lot of sub-headings that create fragmentation of the paper.

The paper has a lot of tables that are difficult to digest, some of them can be put in appendix and inserting in the text the most important ones to improve the readability…

Table 1-2-3-5 are long and can be broken in two, one per page, using as title, table 1, table 1 continue, etc.…

Table 6, R2 is low and this can be due to a mis specified model and likely consequential weak and misleading results. Significance can be indicated with stars, as in econometrics textbook, such as ***=0.001, **=0.05, etc.…

Paper dense of information that create confusion.

Discussion has to be short.

First, authors have to synthesize the main results in a simple table to be clear for readers and then show what this study adds compared to other studies.

Analysis of Findings. Although the Results section provides a detailed description of the data collected, there needs to be a more critical synthesis and comparison of the findings in the discussion.

• Explanation of results: Comment on whether the results were expected for each set of findings; go to greater depth to explain unexpected or incredibly profound findings. If appropriate, note any unusual or unanticipated patterns or trends that emerged from your results and explain their meaning concerning the research problem.

Limitations can be put in conclusion.

Conclusion has not to be a summary, but authors have to focus on manifold limitations of this study and provide implications to support best practices for criminal justice system.

Make sure you create subsections in the Conclusion: 1) Theoretical Implications, 2) Policy Implications, and 3) Ideas for Future Research.

Overall, then, the paper is interesting, but the structure is long and is confused. Theoretical framework can be improved, and some results create confusion… structure of the paper has to be improved; study design, discussion and presentation of results have to be clarified using suggested comments.

I strongly suggest improving the paper, by using all the comments (suggested papers included to read and use all) that I will verify in depth, and maybe it can be considered.

Sure enough , if the paper is not improved as suggested it will be dismissed.

Suggested readings of relevant papers that have to be read and all inserted into the text and references.

Barrantes-Chaves, K. 2024. More than walls: Fear of crime in neighbourhoods with different poverty levels bordering gated communities. The Greater Metropolitan Area of Costa Rica, Cities,154, 105331

Syropoulos, S., Leidner, B., Mercado, E., ... Chekroun, P., Rottman, J. 2024. How safe are we? Introducing the multidimensional model of perceived personal safety Personality and Individual Differences, 224, 112640

Coccia M. 2019. The politics of fear and relationship with the effective level of crime and socioeconomic issues: Empirical analysis of a case study, Journal of Economics Bibliography, Vol 6, No. 4, pp. 357-374, http://dx.doi.org/10.1453/jeb.v6i4.1971

Noble, J., Jardin, A. 2024. Mapping fear of crime: defining methodological orientations.Quality and Quantity, 58(2), pp. 1881–1899

Abreu Rivera, L.F., Piatkowska, S.J. 2024. Immigrant threat, perception of unsafety, and political articulation of immigration and multiculturalism in European countries.European Journal of Criminology, 21(6), pp. 830–858

Coccia M. 2017. A Theory of general causes of violent crime: Homicides, Income inequality and deficiencies of the heat hypothesis and of the model of CLASH, Aggression and Violent Behavior, vol. 37, November-December, pp. 190-200, https://doi.org/10.1016/j.avb.2017.10.005

Uding, C.V., Porter, L.C., Dong, B., Moon, H.R. 2024. Violence, place, and health: A review of the literature, Aggression and Violent Behavior, 78, 101983

Coccia, M., Cohn, E.G. & Kakar, S. How immigration, level of unemployment, and income inequality affect crime in Europe. Crime Law Soc Change (2024). https://doi.org/10.1007/s10611-024-10144-y

Roman, C.G., Chen, R., Natarajan, L., ... Glanz, K., Sallis, J.F. 2024. Crime-related perceptions and walking for recreation inside and outside one's home neighborhood. Health and Place, 89, 103316

Reviewer #2: Please revise the paper as per the following comments:

1. The paper identifies two new respondent classes, but the comparison with previous versions of the EEF model is not fully detailed.

2. The results section discusses variability in sociodemographic predictors but does not provide detailed insights into the significance or effect sizes of these predictors.

3. While the implications for public policy are mentioned, they are not elaborated sufficiently.

4. The paper focuses on a French dataset but does not discuss how the findings might generalize to other cultural or geographical contexts.

5. The results section discusses variability in sociodemographic predictors but does not provide detailed insights into the significance or effect sizes of these predictors.

**Do you want your identity to be public for this peer review?** For information about this choice, including consent withdrawal, please see our Privacy Policy

Reviewer #1: No

Reviewer #2: No

---

## [Author Response · Author response to Decision Letter 1]

6 Mar 2025

Response to Reviewers

1. Structural and Organizational Concerns:

- The authors are encouraged to amend the fragmented and confusing structure of the paper, they are recommended to completely reorganize the manuscript into clearly defined sections (e.g., Introduction, Theoretical Framework, Study Design, Results, Discussion, Conclusion).

The paper has been completely reorganized following the recommendations of the first reviewer (see the responses to the first reviewer's comments).

- The excessive use of subheadings creates unnecessary complexity and redundancy, which detracts from the paper’s readability and coherence. Simplifying and synthesizing sections (e.g., merging theoretical framework and earlier versions of the model) is critical.

The requests for the removal of subtitles have all been followed. In accordance with the recommendations of the first reviewer, we have replaced the subtitles with bullet points. More generally, all proposed structural modifications have been adopted (see the responses to the first reviewer’s comments).

However, regarding the request to summarize the sections, we emphasize that the method used in this article is not deployed elsewhere. It is therefore crucial to explain it thoroughly. This article should be seen as a foundation for further analyses. With this objective in mind, PLOS One appeared to be a particularly suitable choice. Indeed, your journal offers the significant advantage of allowing researchers to provide detailed explanations of their methodologies and results without imposing any word limit. That being said, we have made every effort to be as concise as possible.

2. Methodological Clarity and Rigor:

- The methodology is unclear and requires restructuring into distinct subsections (Sample, Measures, Models, and Data Analysis). Additionally, specific elements like the logistic model equation and a consistent taxonomic criterion for group categorizations need to be clearly explained and justified.

The methodology section has been restructured according to the first reviewer's recommendations. The regression model equation was already present in the first version of the article. However, we have modified the way the equation's terms are presented to improve readability.

Regarding the classification analysis, we have added a section to explain why no taxonomic criterion is defined a priori. We have thus provided a clearer explanation of the inductive approach of the method used (see the responses to the first reviewer's comments).

- Authors should provide sufficient detail on effect sizes and the significance of sociodemographic predictors, which undermines the credibility of the results.

The "Limitations" section has been revised and expanded to provide more detail (see the responses to the first and second reviewers' comments).

3. Presentation of Results:

- The presentation of results is dense and difficult to digest. Tables and figures are either unreadable or redundant (e.g., Table 6 with low R2R^2 and Figure 1). Authors are advised to consolidate results into fewer, more meaningful tables and moving supplementary details to an appendix is necessary.

Since all the presented results are statistically significant, it is difficult to select which ones should remain in the main text and which should be placed in the appendix. One of the solutions suggested by the first reviewer was to subdivide the densest tables into multiple parts. We have therefore adopted this approach (see the responses to the first reviewer’s comments).

- Authors are also recommended to synthesize key findings in a simple table within the discussion for better clarity.

Table 8 has been created to address this request. It summarizes our main results while comparing them to those of previous versions of the EEF model (see the responses to the first reviewer’s comments).

4. Discussion and Implications:

- There is a need for a more critical synthesis and deeper analysis in the discussion. A concise explanation of results, highlighting unexpected or profound findings, and their relevance to existing literature.

The "Discussion" section has been extensively revised to address this comment. In addition to the creation of Table 8, the section has been reorganized, and bullet points have been added to highlight the key contributions of our results compared to previous studies. The hypotheses—integrated into the "Theoretical Framework" section following the first reviewer’s advice—have been discussed in light of the obtained results. Finally, this section now includes the references suggested by the first reviewer (see the responses to the first and second reviewers’ comments).

- Practical and policy implications are underdeveloped, insufficient elaboration on public policy implications and generalizability of findings to other contexts.

Following the recommendations of both reviewers, we have expanded the section on policy implications (see the responses to the first and second reviewers’ comments). Additionally, in the conclusion (section "Ideas for Future Research"), we discuss the possibility of generalizing our findings to other geographical and cultural contexts (see the responses to the second reviewer’s comments).

5. Theoretical Framework:

- Authors are also encouraged to improve the theoretical framework by incorporating relevant suggested readings to provide a stronger foundation for the analysis. Additionally, previous versions of the model should be clearly synthesized into a cohesive narrative.

The references suggested by the first reviewer have been included in the "Discussion" section rather than in the "Theoretical Framework" section. Given their content, we considered that they were more appropriately placed in the discussion.

Additionally, following the first reviewer’s advice, we have created a table (Table 1) that summarizes the key results from previous versions of the EEF model.

- The theoretical contribution of the paper is unclear without these improvements.

Indeed, all these modifications significantly improve the clarity and readability of the paper (see the responses to the first reviewer’s comments).

6. Limitations and Conclusions:

- Authors are advised to place limitations in the conclusion and focus on theoretical, policy, and future research implications. Subsections within the conclusion can improve clarity and emphasize the paper’s contribution to the field.

If the editor and reviewers have no objections, we would prefer to leave the "Limitations" section separate from the "Conclusion." This solution offers two advantages. First, it prevents the conclusion from becoming overly lengthy (which now contains three sub-sections in bullet-point format). Second, it highlights the limitations of a method that is not used elsewhere. The "Limitations" section also aims to reflect on ways to address these issues in future studies (see the responses to the first reviewer’s comments).

7. Scope and Generalizability:

- One of the main concerns is the lack of discussion on the generalizability of findings beyond the French context. Explicitly addressing how the findings may or may not apply to other cultural or geographical contexts will enhance the paper's broader relevance.

This point is addressed at the very end of the article.

Comments from PLOS Editorial Office: We note that one or more reviewers has recommended that you cite specific previously published works. As always, we recommend that you please review and evaluate the requested works to determine whether they are relevant and should be cited. It is not a requirement to cite these works. We appreciate your attention to this request.

Reviewers' comments:

Reviewer's Responses to Questions

Comments to the Author

1. Is the manuscript technically sound, and do the data support the conclusions?

Reviewer #1: Partly

Reviewer #2: Yes

2. Has the statistical analysis been performed appropriately and rigorously?

Reviewer #1: Yes

Reviewer #2: Yes

3. Have the authors made all data underlying the findings in their manuscript fully available?

Reviewer #1: No

Reviewer #2: Yes

4. Is the manuscript presented in an intelligible fashion and written in standard English?

Reviewer #1: Yes

Reviewer #2: Yes

5. Review Comments to the Author

Reviewer #1: Patterns of fear of crime. Revisit and synthesize the model of experiential and expressive fear of crime

The topics of this paper are interesting, though well known. The structure and content must be revised, and results have to be better explained by authors before to be reconsidered.

Title has to be shorter, clarify if this is a case study in France and the period.

We have modified the title of the article. Instead of " Patterns of fear of crime. Revisit and synthesize the model of experiential and expressive fear of crime," we propose " Patterns of fear of crime: A mixed-methods exploration of the model of experiential and expressive fear of crime." This title emphasizes the originality of the paper, which is based on two statistical analysis methods, the combination of which is rarely used in the study of fear of crime.

Regarding the inclusion of the country in which the study was conducted, we would prefer not to mention it in the title. Aside from the fact that this information would further lengthen the title, we believe it could reduce the interest in the article by suggesting, from the title, that the results would only apply to the case of France. As the paper’s focus is on identifying the structures of relationships between variables (rather than, for example, presenting prevalences), we believe the results can extend beyond the French context.

Abstract has to be shorter, clarify the goal, what the study adds and implications for criminal justice system. Also here clarify if data are related to France and period under study.

We have revised the abstract following the reviewer’s suggestions. The contributions of the study to 1) research and 2) the criminal justice system have been reformulated, while adhering to the word limit imposed by the journal (300 words).

The paper is long and has to be reduced. The structure is too fragmentated and confusing.

The reviewer details these points in the following comments. We will address our modifications as we go through their remarks.

Authors have to structure the paper as follows.

-Introduction

-Theoretical framework

-Proposed model and Study design

-Results

-Discussion

-Conclusion

Authors have to avoid subheadings that create fragmentation and confusion. If necessary, can use bullet points (same comments for all sections).

We have adopted the structure suggested by the reviewer. We have removed the subtitles and replaced them with bullet points.

Introduction has to better clarify the research questions of this study and provide more theoretical background about these topics, the socioeconomic drivers of crime in society, etc. After that they can focus on the topics of this study to provide a correct analysis for fruitful discussion. (See suggested readings that must be all read and used in the text).

As requested, the research question has been explicitly stated. We have also revised the wording of the article's objectives. Two bullet points have been added to highlight these objectives.

However, to avoid overloading the text and to prevent repetition with the subsequent sections, we prefer to incorporate the suggested works into the main body of the text at appropriate points, rather than strengthening the theoretical background in the introduction. Indeed, the theoretical background for the EEF model is quite extensive. As mentioned in the introduction, this unifying model "integrates, from an updated literature review, the 'factors' recognized by previous studies as being the most contributive to this social phenomenon." These factors are numerous, and we intentionally limit the introduction to this description to avoid repetition with the next section (theoretical framework), which goes into detail about the origins of the model.

Sections From the model's origins to its design, The results of the first version of the model, The results of the second version of the model, The results of the third version of the model can be merged and called “theoretical framework” reducing the content. A figure can synthetize these results of the previous version of the model to be clear for readers.

The "Theoretical Framework" section has been created. The sections "From the model's origins to its design," "The results of the first version of the model," "The results of the second version of the model," and "The results of the third version of the model" have been integrated into the "Theoretical Framework" section and are now presented in bullet points.

Table 1, titled "Comparative Synthesis of the Three Versions of the EEF Model," has been added at the end of the section. As requested, it summarizes the main methodological characteristics and key results of the three versions of the EEF model.

Section Hypothetico-Deductive Reasoning to Inductive Reasoning can be included in the study design. Reduce the section and provide a clear working hypothesis alternative or complementary to previous approaches.

The content of the section "Hypothetico-Deductive Reasoning to Inductive Reasoning" has been integrated at the beginning of the "Proposed Model and Study Design" section. We have placed a footnote (85) regarding the difficulty for an SEM model to identify multiple relationships between variables. This helps significantly reduce the content of the section.

Additionally, we have explicitly written our three working hypotheses, which are placed at the end of the section in bullet points. Finally, to reduce the confusion mentioned by the reviewer, the subtitle "Hypothetico-Deductive Reasoning to Inductive Reasoning" has been removed.

The methods of this study is not clear. The section of Materials and methods must be re-structured with the following three sections and same order:

• Sample and data

• Measures of variables

• Models and Data analysis procedure. Logistic model has to be specified with equation.

Authors, again, have to avoid subheadings that create fragmentation and confusion. If necessary, you can use bullet points (same comments for section of results and all sections). A flow chart can show the logic of reasoning, indicating expected results (+ or -) to support then with empirical evidence and methods of inquiry to be clear for readers.

Categorization of gro

---

## [Decision Letter · Decision Letter 1]

16 Jul 2025

Patterns of fear of crime. A mixed-methods exploration of the model of experiential and expressive fear of crime

PONE-D-24-43257R1

Dear Dr. Noble,

We’re pleased to inform you that your manuscript has been judged scientifically suitable for publication and will be formally accepted for publication once it meets all outstanding technical requirements.

Kind regards,

Wing Hong Chui, PhD

Academic Editor

PLOS ONE

Additional Editor Comments (optional):

Reviewers' comments:

Reviewer's Responses to Questions

**Comments to the Author**

Reviewer #1: All comments have been addressed

Reviewer #2: (No Response)

2. Is the manuscript technically sound, and do the data support the conclusions?

Reviewer #1: Yes

Reviewer #2: (No Response)

3. Has the statistical analysis been performed appropriately and rigorously?

Reviewer #1: Yes

Reviewer #2: (No Response)

4. Have the authors made all data underlying the findings in their manuscript fully available?

Reviewer #1: Yes

Reviewer #2: (No Response)

5. Is the manuscript presented in an intelligible fashion and written in standard English?

Reviewer #1: Yes

Reviewer #2: (No Response)

Reviewer #1: I have read thoroughly the revised version of paper.

Now this version of the paper after revision done is OK and provides interesting results for readers.

Reviewer #2: The authors have revised their paper as per all my comments from the previous review round. I do not have any other comments at this point. I recommend the publication of this paper in its current form.

**Do you want your identity to be public for this peer review?** For information about this choice, including consent withdrawal, please see our Privacy Policy

Reviewer #1: No

Reviewer #2: No

---

## [Editor Report · Acceptance letter]

PONE-D-24-43257R1

PLOS ONE

Dear Dr. Noble,

I'm pleased to inform you that your manuscript has been deemed suitable for publication in PLOS ONE. Congratulations! Your manuscript is now being handed over to our production team.

Kind regards,

on behalf of

Prof. Wing Hong Chui

Academic Editor

PLOS ONE